# Enhancing the diagnosis of functionally relevant coronary artery disease with machine learning

Christian Bock [1,2,8], Joan Elias Walter[3,4,5,8], Bastian Rieck [1,2,6,8], Ivo Strebel[3,4], Klara Rumora[3,4], Ibrahim Schaefer[3,4], Michael J. Zellweger[3,4], Karsten Borgwardt [1,2,7,9] ✉ & Christian Müller [3,4,9] ✉

Functionally relevant coronary artery disease (fCAD) can result in premature death or nonfatal acute myocardial infarction. Its early detection is a fundamentally important task in medicine. Classical detection approaches suffer from limited diagnostic accuracy or expose patients to possibly harmful radiation. Here we show how machine learning (ML) can outperform cardiologists in predicting the presence of stress-induced fCAD in terms of area under the receiver operating characteristic (AUROC: 0.71 vs. 0.64, $p = 4.0\text{E-}13$). We present two ML approaches, the first using eight static clinical variables, whereas the second leverages electrocardiogram signals from exercise stress testing. At a target post-test probability for fCAD of <15%, ML facilitates a potential reduction of imaging procedures by 15–17% compared to the cardiologist's judgement. Predictive performance is validated on an internal temporal data split as well as externally. We also show that combining clinical judgement with conventional ML and deep learning using logistic regression results in a mean AUROC of 0.74.

Coronary artery disease (CAD) is the leading cause of death worldwide[1–3]. High mortality and morbidity rates, paired with the availability of highly effective and cost-efficient prevention and treatment measures, underline the importance of early risk stratification of patients with suspected CAD. CAD may be clinically silent for decades or become functionally relevant (fCAD) by causing symptoms of myocardial ischaemia that impact the quality of life and potentially result in significant or adverse cardiac events such as premature death or nonfatal acute myocardial infarction (AMI) in the further course. Therefore, detection strategies should focus on fCAD to maximise patient benefit. Unfortunately, rapid, easy, and safe rule-out of fCAD

remains a major unmet clinical need. The practical utility of current screening techniques is limited by either unfavourable diagnostic accuracy, as in the case of exercise electrocardiography stress testing, or by their obtrusive nature and high costs, as in the case of functional non-invasive imaging such as myocardial perfusion imaging (MPI) or anatomical non-invasive evaluation such as coronary computed tomography angiography[4–9]. While these dedicated cardiac imaging techniques can benefit many patients, they appear to be increasingly employed improperly in patients with a low pre-test probability of fCAD[5–7]. Considering the large population at risk, as well as the available prevention and treatment options, a clinical tool that enables

[1]Department of Biosystems Science and Engineering, ETH Zürich, Basel, Switzerland. [2]Swiss Institute for Bioinformatics, Lausanne, Switzerland. [3]Cardiovascular Research Institute Basel, University Hospital of Basel, University of Basel, Basel, Switzerland. [4]Department of Cardiology, University Hospital of Basel, University of Basel, Basel, Switzerland. [5]Department of Endocrinology, Diabetology and Clinical Nutrition, University Hospital Zurich, University of Zurich, Zurich, Switzerland. [6]Institute of AI for Health, Helmholtz Munich and Technical University of Munich, Munich, Germany. [7]Department of Machine Learning and Systems Biology, Max Planck Institute of Biochemistry, Martinsried, Germany. [8]These authors contributed equally: Christian Bock, Joan Elias Walter, Bastian Rieck. [9]These authors jointly supervised this work: Karsten Borgwardt, Christian Müller. ✉e-mail: borgwardt@biochem.mpg.de; christian.mueller@usb.ch

effective, efficient, and safe detection of fCAD can improve patient outcomes while reducing the burden on patients as well as health care costs.

The automated detection of cardiac events has a long history[10,11], and traditionally employed methods rely on quantifying ECG changes such as ST-segment elevation/depression, T-wave abnormalities, or other morphological anomalies of the QRS complex. However, a significant drawback of these methods is their reliance on ECG delineation algorithms that locate the segments a heartbeat is composed of. Delineation results can be inaccurate[10] for abnormal heartbeats, thus substantially limiting their utility for at-risk patients. Over the last few years, deep learning (DL) emerged as a powerful tool to build classification systems from ECG signals that do not require engineering QRS complex features[12,13]. Particularly in detecting different cardiac arrhythmias, the classification performance of DL systems reached the point of cardiologist-level accuracy[14,15]. While the potential of DL has been investigated in the context of cardiac stress testing[16–20], previous work has the following drawbacks: (1) usage of a large number of variables which exacerbates model transferability, (2) reliance on summary statistics computed from automated ECG delineation or automated and less accurate outcome definitions, (3) lack of comprehensive performance evaluations on diverse subcohorts, and (4) lack of external validation. Lastly, ours is the first study investigating the benefit of collaborative machine learning in predicting abnormal myocardial perfusion.

Recent cardiology clinical practice guidelines[8,21] discouraged the sole use of stress ECG testing due to low diagnostic accuracy and unacceptable false negative and positive rates. However, given its wide availability, ease of use, and low cost, stress ECG testing remains commonly performed, which demands methods to use data acquired during stress testing more effectively. In addition, a stress ECG contains a plethora of information that cannot be included in routine clinical assessment (such as subtle morphological changes over time) but can serve as clinically relevant input for a DL system. At the same time, conventional machine learning based on static clinical variables alone has been shown to be at least as powerful as more complex deep neural networks in the healthcare setting[22–24].

Thus, our aim is to derive and validate two different machine learning models in a heterogeneous patient population with a wide range of pre-test probabilities, namely (1) an ensemble learning model based on basic available clinical information, and (2) a deep learning model based on the aforementioned non-sequential variables as well as the ECG signals obtained during stress testing. We compare these models to the clinical assessment of cardiologists after stress testing. Furthermore, to extend their possible scope, the models were also trained and evaluated in patients who are usually excluded from stress ECGs and compared with the cardiologist's clinical assessment after pharmacological testing.

## Results
### Data collection, label generation, and robustness
Panel a of Fig. 1 illustrates our data generation workflow. We collected stress test ECG data from 3522 consecutive adult patients who underwent a standard[25] rest/stress myocardial perfusion single-photon emission computed tomography (SPECT) protocol at a tertiary hospital as part of the BASEL VIII study (NCT01838148). Patients were referred with symptoms possibly related to inducible myocardial ischaemia and clinical suspicion of stable coronary heart disease. If a patient was not able to reach their target heart rate, a pharmacological protocol with either adenosine or dobutamine was initiated by the treating clinician. Individuals for whom stress test by bicycle ergometry was not possible were put on a pharmacological protocol from the start. To compare the algorithmic approaches with expert judgement, the treating cardiologist performed a clinical assessment before and after stress testing: considering all available medical information

such as (cardiac) history, relevant symptoms, risk factors, (stress) ECG, prior imaging and more, they indicated the probability of the presence of fCAD on a visual analogue scale (VAS) from 0% to 100%[26–29]. Representing clinical practice, adjudication of functionally relevant CAD was not formally blinded for stress ECG results or demographics and was performed centrally by an expert team composed of a nuclear medicine physician and a cardiologist assessing myocardial perfusion scans. Furthermore, whenever available, adjudication was refined with coronary angiography and fractional flow reserve assessment. Of the 3522 eligible patients who provided written informed consent, 701 (20%) patients underwent coronary angiography within 3 months, with 30 (0.9%) patients being reclassified to the fCAD group and 74 (2.1%) being reclassified to the non-ischaemic group. The VAS score the treating cardiologist provides after the stress test but before they get access to the imaging results represents the cardiologist baseline in our study. In practice, this can be interpreted as an indicator as to whether the cardiologist would recommend a follow-up examination with advanced imaging.

The data set was split into a development (75%) and a held-out test set (25%). All patients in the development set enrolled in the study from Jan. 2010 through Dec. 2014; the held-out test set contains patients who enrolled from Dec. 2014 through May 2016. It was only released and used once the models' parameters were fixed. Thus, high predictive performance on the held-out test set indicates the robustness of our system's generalisation capability with respect to a temporal shift of the data[30], paving the path towards subsequent real-world applications. Lastly, we use external data from two Israeli medical centres to validate our system on 916 consecutive patients referred for SPECT MPI testing, whose ECG signals were obtained by treadmill stress test. This evaluation scenario is designed to exemplify the ability of both computational approaches to generalise to patients from unseen institutions, new modalities, and highlight their behaviour under distributional shifts. Given an fCAD prevalence of 7.5% in the external data set, our approach based on clinical data alone (AUROC: $0.75 \pm 0.004$, AUPRC: $0.19 \pm 0.01$) is outperformed by our deep neural net using ECG time series and clinical variables (AUROC: $0.80 \pm 0.01$, AUPRC: $0.28 \pm 0.01$). Please refer to the Method section for more details on data splitting and distributional shifts in the external validation data.

### Development of an ensemble predictor and a multi-task neural network for functionally relevant CAD prediction
The ability to learn from raw sequential data (i.e., time series) makes neural networks a popular approach for healthcare applications. However, conventional machine learning (ML) has shown to be at least as powerful as deep learning in the clinical context[22,23], thus creating opportunities for low-cost deployments that do not require specialised hardware. Therefore, we will also compare their performance to deep learning models. To this end, we select a small set of eight non-sequential, easy-to-access variables on which we train four conventional ML methods (i.e., decision trees, random forests[31], logistic regression, support vector machines[32]). These variables include age, weight, biological sex, height, heart rate at rest, systolic and diastolic blood pressure, and presence of a previous CAD. The best-performing approach (a random forest) was selected via 5-fold cross-validation. We refer to all developed predictors as Coronary ARtery disease PrEdictor (CARPE). Based exclusively on clinical data, we refer to the random forest model as CARPE_{Clin.}. Additionally, we develop a neural network approach, CARPE_{ECG}, that uses the aforementioned non-sequential variables and the ECG signal, as illustrated in panel c of Fig. 1. We trained CARPE_{ECG} via a multi-task learning[33] (MTL) architecture with residual layers[15,34] at its core using the torchmtl[35] package. MTL uses so-called auxiliary tasks (i.e., prediction targets) related to a main task (e.g., fCAD prediction). These domain-specific inductive biases ensure improved and robust predictive performance on the

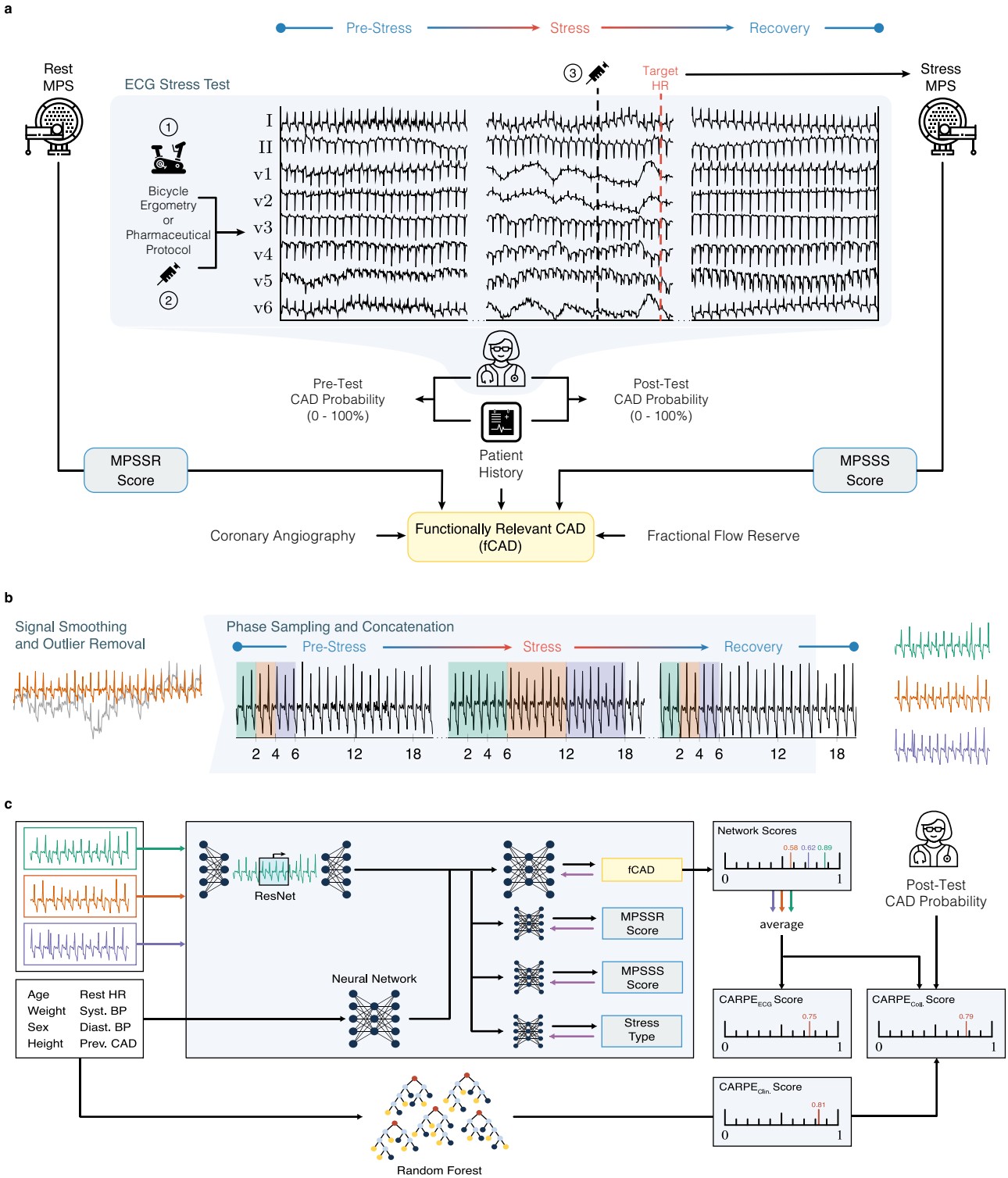

main task[36]. As shown in Fig. 1, we train CARPE_ECG on three auxiliary tasks (blue boxes), two of which (MPSSS and MPSRS) quantify the heart's perfusion capabilities without and under stress, respectively. The third auxiliary task is to predict whether a patient received any pharmacological support to perform the stress test. Each auxiliary task impacts performance on the main task differently. Their respective importance weights were selected in a grid search on the three best-performing leads (see Supplementary Fig. 5 and Supplementary Table 5). To gain insights into the importance of static features and ECG segments, we used SHAP (SHapley[37] Additive exPlanation) values[38].

Finally, we combine predictions from the ensemble model and deep learning approach with the cardiologist's post-test judgement by training a new logistic regression model on all three scores from the training set. This way, we leverage the experience and domain knowledge of the cardiologist while adding the potential to benefit from supervised learning techniques. We believe that, in practice, such a collaborative approach has the highest chances of being accepted in a clinical setting not only because it reaches the highest diagnostic performance (see Supplementary Table 6) but also because the cardiologist is an integral part of the score generation (in fact they are required to provide a VAS score after stress testing). Nevertheless, the

**Fig. 1 | Protocol overview. a** Data acquisition: We highlight the three primary subgroups of exercise stress testing: ① patients who complete the bicycle exercise stress test, ② patients not able to exercise on the bicycle, and for whom a pharmaceutical protocol is used at the beginning of the stress test, and ③ patients starting on the bicycle but need pharmacological support to reach their target heart rate. Doctors perform myocardial perfusion scans at rest (rest MPS), and at the target heart rate (stress MPS). Myocardial perfusion is quantified by the myocardial perfusion scan summed rest score (MPSSR score), and the MPS summed stress score (MPSSS score). The cardiologist estimates the probability of a functionally relevant CAD (fCAD) before and after the stress test (Pre/Post-Test CAD Probability in the figure). The binary label indicating the presence of fCAD (yellow box) is adjudicated by considering the stress test results and additional relevant clinical parameters. **b** Data Preprocessing: Following smoothing and outlier removal, time series that serve as input to the neural network are constructed by joining short subsequences from different phases of the stress test. For this, 2 s from the pre-stress phase, 6 s from the stress phase, and 2 s from the recovery phase are sampled and concatenated multiple times for a single patient (green, orange, and purple sequences). $x$-axes represent time in seconds. **c** Machine Learning: For our neural network approach (CARPE$_{ECG}$), these 2-6-2 sequences are fed into a residual neural network (ResNet). In parallel, the patient's static clinical data are processed by a 2-layer feedforward network. Four subnetworks are trained on three auxiliary tasks (i.e., MPSSR & MPSSS score as well as stress type prediction) and one main task (fCAD prediction). We average predictions of the main task over all 2-6-2 sequences per patient. Purple arrows in front of each task indicate the direction of the learning signal. The same clinical variables as for CARPE$_{ECG}$ are used to train a random forest classifier (CARPE$_{Clin.}$); nodes are coloured to enhance legibility. We combine both predictions with the cardiologist's judgement in a logistic regression model (CARPE$_{Coll.}$).

use of logistic regression to enhance diagnostic accuracy does not ensure or directly translate to clinical utility. The precise impact on patient risk stratification needs to be assessed separately.

## Machine learning can be used to reduce unnecessary perfusion imaging

The prevalence of centrally adjudicated fCAD was 32.9% in the full study cohort and 28% in the held-out test split. Figure 2 depicts the diagnostic performance of our machine learning approaches, the cardiologist's assessment after stress testing, and a computational approach that uses the ECG's ST-segment depression[39,40] as an indication of the presence of fCAD (see Methods for a detailed description) on the held-out data set. We show receiver operating characteristic (ROC) and precision-recall curves in the first row. Standard deviations shown as envelopes were obtained using bootstrapping, as detailed in the Methods section. Regarding the mean area under the ROC curve, we observe that CARPE$_{ECG}$ (0.71) and CARPE$_{Clin.}$ (0.70) outperform both the ST-depression algorithm (0.58) and the cardiologist (0.64). In regions of high specificity, CARPE$_{Clin.}$ drops below the sensitivity of the cardiologist, while CARPE$_{ECG}$ reaches comparable predictive performance (see inline plot). At the other extreme of the ROC curve, i.e., at high sensitivity values, both machine learning approaches consistently lead to a higher specificity than the cardiologist's judgement (see inline plot).

Decision curves[41] (rows two and three in Fig. 2) overcome the drawbacks of conventional performance evaluations and calibration analyses[42] by focusing on a predictor's clinical value. The concept of net benefit quantifies the trade-off between diagnosing sick patients and preventing healthy patients from being exposed to harmful testing procedures[43]. For a specific decision threshold probability of a diagnostic tool, a larger net benefit indicates a greater number of true positive predictions without an increase in the rate of false positives and, conversely, a greater number of true negative predictions without an increase in false negatives. Figure 2 shows a decision curve analysis in the second and third row with pre-test rule-out cutoffs (dotted red) as advocated for in European and US-American guidelines[8,21], which consider probability thresholds between 5-15% for further non-invasive imaging. The European guideline, for instance, considers non-invasive testing in patients with a probability >15% as most beneficial and testing in patients with 5–15% as potentially beneficial. Our machine learning models lead to a higher net benefit than the cardiologist's assessment at all thresholds. Notably, at the threshold of 15%, relying on the cardiologist's judgement is worse (in terms of net benefit) than performing myocardial perfusion imaging on all patients demonstrating the value of an ML-based method.

Table 1 offers a detailed decision curve analysis, showing sensitivity, negative predictive value (NPV), and percentage of avoided myocardial perfusion imaging compared to the cardiologist's judgement at three probability cut-off values. We also show the percentage of patients who received a score below the cutoff threshold to enable a meaningful interpretation of sensitivity values. The highest fraction of

MPIs, i.e., almost 25 per 100 patients, could be avoided at a decision threshold of 10% by using CARPE$_{Coll.}$ as a risk stratification method due to risk-overestimation by the cardiologist. That being said, cutoff thresholds should not be chosen to optimise diagnostic performance, but they represent the cardiologist's minimum probability of disease at which an intervention would be warranted[42]. In other words, if a cardiologist holds the belief that missing a patient who suffers from fCAD is nine times worse than performing an unnecessary MPI, a model's performance should be assessed at the 10% cutoff.

Evaluating CARPE$_{ECG}$ as a predictive model on all patients of the held-out test set (at the 15% decision threshold) shows the potential to reduce perfusion imaging by 15.3% (see Table 1) without increasing the rate of false negatives. This number increases to 17.3% when using CARPE$_{Coll.}$. We observe similar behaviour in patients without a CAD history. At the 5% threshold (i.e., if a physician considers it 19 times worse to miss an fCAD diagnosis than to perform an unwarranted MPI), ML can be used to avoid 10.8% of the imaging ordered by a cardiologist. For patients with CAD history, the decision thresholds of 5% and 10% lead to a particularly small number (<1% or none) of patients for which fCAD can be ruled out, which inflates sensitivity and NPV of CARPE$_{ECG}$ and CARPE$_{Coll.}$. This inflation is particularly pronounced in CARPE$_{Clin.}$ (see Supplementary Fig. 6) which is therefore not shown here. Overall, these results demonstrate the potential clinical utility of the proposed methods to reduce potentially unwarranted MPIs.

## Subcohort analysis: machine learning performs particularly well on younger patients

Trustworthiness and interpretability are of fundamental importance in the development of risk stratification models in cardiology[44]. Identifying (sub)cohorts of the population for which the model performs particularly well or poorly is crucial. To address the issue of trust, we evaluate our models' performances on a variety of subcohorts that are important in the context of exercise stress testing. Regarding interpretability, we perform an analysis of SHAP values[38] on the population level, and a case study to better understand the impact feature values and ECG segments have on the predicted scores.

Clinically significant subgroups include patients who underwent exercise stress testing versus patients who required pharmacological testing as well as patients without a prior history of CAD versus patients with a known history of CAD; the odds of suffering from fCAD are significantly increased ($p = 2.26E-40$, two-sided Fisher's exact test, test statistic = 2.64) for patients with previous CAD (OR: 2.64, 95% CI: 2.28–3.05) over the whole cohort. To obtain a more detailed performance breakdown, we also stratify the data by sex and age. Diagnostic performances of all approaches and subcohorts of the held-out test set are shown in Fig. 3 and Supplementary Table 6. For comparison, the performance of the CAD consortium model[45] and the currently used ESC pre-test probabilities for obstructive coronary artery disease[8,9], both based on age, sex, and the nature of symptoms, is shown in patients without known coronary artery disease. First, we assess the

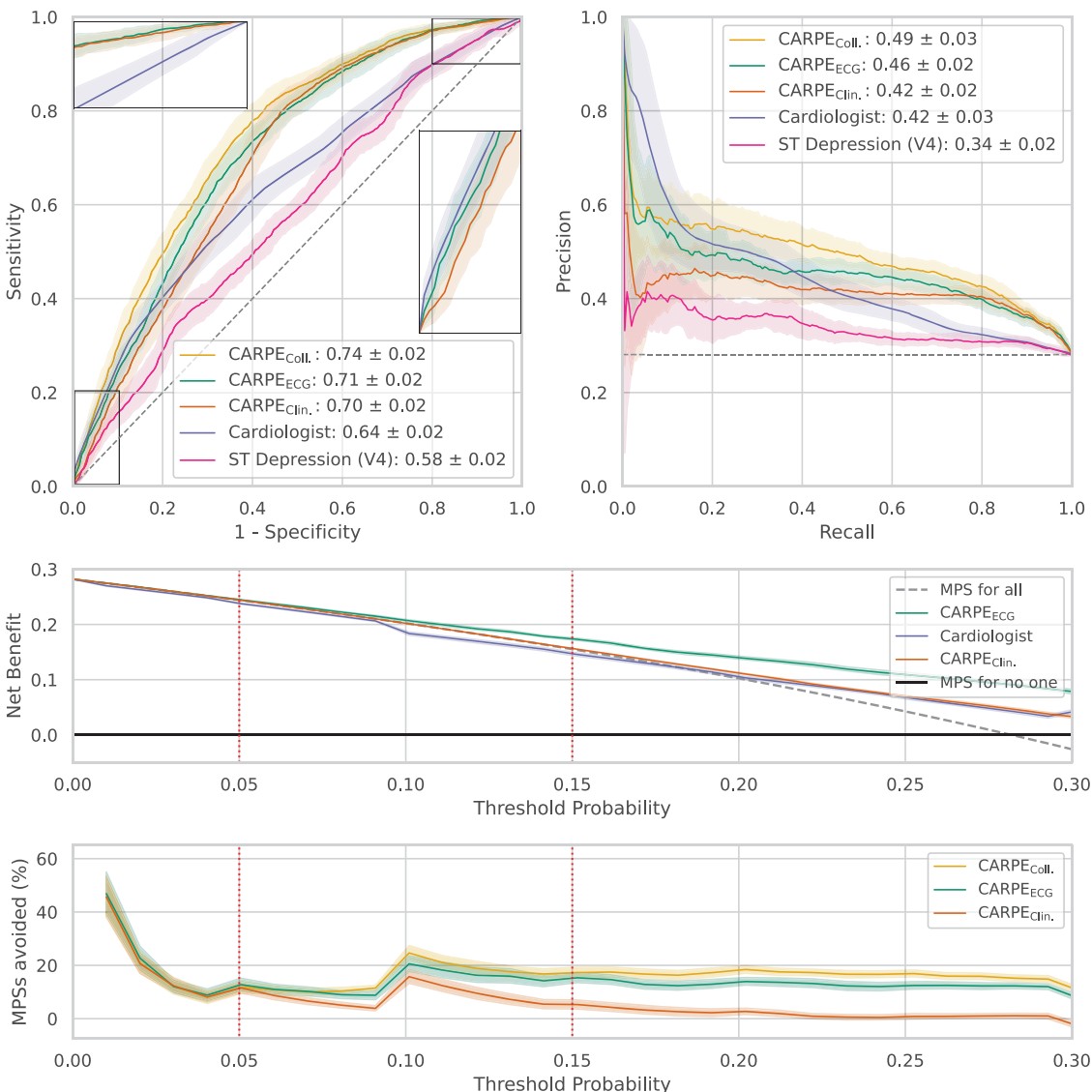

**Fig. 2 | Diagnostic performance overview.** ROC and PR-curve. Predictive performance of our deep learning-based approach (CARPE$_{ECG}$), a random forest based on clinical data (CARPE$_{Clin.}$), the cardiologist, and ST depression in terms of mean performance ± standard deviation (envelopes) over $n = 25$ bootstrap draws. The upper plots show that both machine learning approaches outperform the cardiologist in terms of area under the receiver operating characteristic and precision-recall curve. In regions of high specificity (inline plot), the neural network is on par with the cardiologist while CARPE$_{Clin.}$ exhibits worse performance. Both machine learning methods outperform the cardiologist's judgement in regions of high sensitivity (inline plot). Decision Curve: First row: Net benefit[43] plot for CARPE$_{ECG}$ (green), CARPE$_{Clin.}$ (orange), the cardiologist (purple), a myocardial perfusion scan (MPS) for no patient (black), and MPS for all patients (dashed grey). CARPE$_{Coll.}$ is not shown as it is visually indistinguishable from CARPE$_{ECG}$. Net benefit puts both benefit and harm on the same scale. In our case, we consider harm to be inflicted by performing an unnecessary MPS. At a decision threshold of 5%, all approaches lead to a similar net benefit. At the second threshold of 15%, CARPE$_{Clin.}$ and the cardiologist demonstrate a net benefit similar to performing MPS on all patients, with CARPE$_{ECG}$ leading to a higher net benefit. Second row: Potential MPSs avoided compared to the cardiologist's strategy: While the conventional ML model and deep learning avoid the approximately same number of MPSs at the decision threshold of 5% (11.5% and 12.8%, respectively), the gap increases at the pre-MPS threshold of 15% (15.3% and 5.3%, respectively). Envelopes in both rows show 95% confidence intervals around the mean over $n = 25$ bootstrap draws. Source data are provided as a Source Data file.

performance of individual machine learning methods before discussing their combination with the cardiologist's judgement. Deep learning outperforms the cardiologist in terms of both AUROC (significant performance increase in 6/10 subcohorts) and AUPRC (significant performance increase in 4/10 subcohorts), while CARPE$_{Clin.}$ exceeds the human baseline in 5/10 strata in terms of AUROC and 2/10 subcohorts in terms of AUPRC. The central plot in panel a of Fig. 4 helps explain this performance discrepancy: The conventional ML model relies more than the neural network on the CAD history and sex variable as visually observable by the large gap between the highest negative and the lowest positive SHAP value for each variable. Strong reliance on a given variable pushes the predictor too strongly in one

direction such that other features cannot compensate for this influence on the final score. This SHAP analysis confirms the importance of the "CAD history", "sex", and "age" variables as observed in other studies[19,20].

Overall, the discriminative performance was highest (excluding CARPE$_{Coll.}$) in younger patients (CARPE$_{ECG}$ AUROC: 0.78 ± 0.04) in general and in younger patients who did not require pharmacological support specifically (CARPE$_{ECG}$ AUROC: 0.79 ± 0.04). The former cohort also represents the stratum in which the increase over the cardiologist is the highest, namely 0.19 in AUROC and 0.15 in AUPRC. We hypothesise that similar to the conventional ML model (i.e., a random forest), a cardiologist might be more biassed towards a

**Table 1 | Detailed Decision Curve Analysis**

| | Method | % below cutoff ± STD | Sensitivity ± STD | NPV ± STD | Myocardial perfusion imaging avoided % (95% CI) |
|---|---|---|---|---|---|
| **Threshold <5% for rule-out** | | | | | |
| All patients (n = 803) | CARPE_ECG | 10.3 ± 1.5 | 0.98 ± 0.01 | **0.96 ± 0.02** | 12.8 (0.4–25.1) |
| | CARPE_Coll. | 5.1 ± 0.9 | **0.99 ± 0.01** | 0.95 ± 0.03 | 11.7 (0.1–23.4) |
| | Cardiologist | 4.3 ± 0.8 | 0.98 ± 0.01 | 0.88 ± 0.05 | baseline |
| No prior CAD (n = 446) | CARPE_ECG | 18.3 ± 2.8 | 0.96 ± 0.02 | **0.96 ± 0.02** | 12.4 (-9.6–34.5) |
| | CARPE_Coll. | 9.1 ± 1.6 | **0.98 ± 0.02** | 0.95 ± 0.03 | 10.8 (-6.8–28.3) |
| | Cardiologist | 4.7 ± 1.1 | **0.98 ± 0.01** | 0.91 ± 0.05 | baseline |
| Prior CAD (n = 357) | CARPE_ECG | 0.07 ± 0.1* | **1.0 ± 0.0** | **1.0 ± 0.0** | 13.1 (-1.3–27.5) |
| | CARPE_Coll. | None | **1.0 ± 0.0** | - | 13.0 (-1.2–27.3) |
| | Cardiologist | 3.8 ± 1.3 | 0.98 ± 0.01 | 0.83 ± 0.12 | baseline |
| **Threshold <10% for rule-out** | | | | | |
| All patients (n = 803) | CARPE_ECG | 21.5 ± 1.3 | 0.94 ± 0.01 | 0.92 ± 0.02 | 20.6 (5.4–35.7) |
| | CARPE_Coll. | 20.3 ± 1.5 | 0.96 ± 0.01 | **0.94 ± 0.02** | 24.6 (11.1–38.1) |
| | Cardiologist | 5.7 ± 0.9 | **0.97 ± 0.01** | 0.85 ± 0.04 | baseline |
| No prior CAD (n = 446) | CARPE_ECG | 37.4 ± 2.0 | 0.86 ± 0.04 | 0.92 ± 0.02 | 12.0 (-7.3–31.3) |
| | CARPE_Coll. | 35.5 ± 3.0 | 0.90 ± 0.03 | **0.94 ± 0.02** | 18.1 (3.2–33.0) |
| | Cardiologist | 5.9 ± 1.1 | **0.96 ± 0.02** | 0.86 ± 0.06 | baseline |
| Prior CAD (n = 357) | CARPE_ECG | 1.0 ± 0.7 | **1.0 ± 0.01** | **0.85 ± 0.23** | 31.4 (11.7–51.1) |
| | CARPE_Coll. | 0.8 ± 0.6* | 1.0 ± 0.0 | 1.0 ± 0.0 | 32.9 (14.7–51.0) |
| | Cardiologist | 5.5 ± 1.3 | 0.98 ± 0.01 | 0.83 ± 0.06 | baseline |
| **Threshold <15% for rule-out** | | | | | |
| All patients (n = 803) | CARPE_ECG | 31.1 ± 1.6 | **0.89 ± 0.02** | 0.90 ± 0.01 | 15.3 (5.4–25.3) |
| | CARPE_Coll. | 32.6 ± 1.8 | **0.89 ± 0.02** | **0.91 ± 0.01** | 17.3 (7.4–27.1) |
| | Cardiologist | 21.5 ± 1.2 | 0.87 ± 0.03 | 0.83 ± 0.03 | baseline |
| No prior CAD (n = 446) | CARPE_ECG | 51.8 ± 2.6 | 0.75 ± 0.05 | 0.90 ± 0.02 | 13.8 (-1.9–29.4) |
| | CARPE_Coll. | 54.9 ± 2.5 | 0.75 ± 0.05 | **0.91 ± 0.01** | 16.7 (3.1–30.4) |
| | Cardiologist | 23.3 ± 2.0 | **0.87 ± 0.03** | 0.89 ± 0.03 | baseline |
| Prior CAD (n = 357) | CARPE_ECG | 4.6 ± 1.3 | 0.98 ± 0.01 | 0.86 ± 0.11 | 17.3 (3.9–30.6) |
| | CARPE_Coll. | 4.0 ± 1.5 | **0.99 ± 0.01** | **0.89 ± 0.10** | 17.8 (5.2–30.4) |
| | Cardiologist | 19.1 ± 2.0 | 0.86 ± 0.03 | 0.73 ± 0.05 | baseline |

Clinically relevant performance indicators for the pre-test probability thresholds of 5%, 10%, and 15%, stratified by presence/absence of CAD history. The column "Myocardial perfusion imaging avoided" uses the cardiologist as the baseline. An asterisk annotates settings with an extremely low number of patients (<1%) who fall below the threshold, leading to an inflation of sensitivity and NPV. Both CARPE_ECG and CARPE_Coll. perform particularly well on patients with a history of CAD at the 15% threshold. On average, we can expect a reduction of unnecessary imaging by 17.8% (CARPE_Coll.) in this cohort at an average sensitivity of 0.99 and a negative predictive value of 0.89. Highest mean values are highlighted in bold.

negative diagnosis in younger patients. In contrast, the DL model is more robust to such a behaviour (as shown by the SHAP distribution of the age variable in Fig. 4). We show a more detailed assessment of diagnostic performance in different age groups in Fig. 5.

On the male subpopulation, CARPE_Clin. is outperformed by the cardiologist, indicating that the conventional ML model relies too strongly on the sex feature as an indicator for the presence of fCAD, whereas the DL model and the cardiologist use this feature more effectively. This is underlined by the observation that the performance gap between CARPE_ECG and CARPE_Clin. is highest in the male subgroup. In female patients, both CARPE_Clin. and CARPE_ECG perform comparably and better than the cardiologist in terms of AUROC.

In patients of at least 65 years of age, it becomes apparent that human judgement and ML might benefit from each other: while individually, both CARPE_Clin. and CARPE_ECG perform equally or worse than the cardiologist, combining all predictions in CARPE_Coll. results in a statistically significant performance increase over the DL model. It appears that the ML models' biases are mitigated by the cardiologist's expertise and vice versa. Augmenting the machine and deep learning output by the cardiologist's judgement also increases diagnostic performance significantly in the full population. While CARPE_Coll. obtains its maximal mean AUROC in the same cohorts as CARPE_ECG, the highest mean increase over CARPE_ECG can be observed in patients with a CAD

history, making it the group in which ML and cardiologists could complement each other most effectively.

## Conventional machine learning relies on age, and ST-segment depressions contribute to high risk scores

For the cardiologist who interacts with a risk-stratification tool, it is critical to understand the model's operations[46] and whether it is consistent with the clinical knowledge about the phenotype. To develop such an understanding, post-hoc explanations[47] can be used to make predictions more interpretable. We use SHAP[38] values, a game-theoretic approach, to explain the outputs of machine learning models. SHAP values provide a score that quantifies the impact an individual feature value has on the model's prediction. A positive SHAP value is associated with the prediction of the positive class/the presence of fCAD. Conversely, a feature with a negative SHAP value influences the model towards predicting the negative class/the absence of fCAD.

Panel a of Fig. 4 shows mean absolute SHAP values and SHAP value distributions for all clinical variables for CARPE_ECG and CARPE_Clin. on the left-hand side. On the right-hand side, we show the SHAP values for the "age" feature. For both classifiers, "CAD history" and "sex" are the most influential predictive features (i.e., highest mean absolute value). However, CAD history is only significantly more relevant than the patient's "sex" in the random forest (p = 7.9E-05, test statistic = 7.36

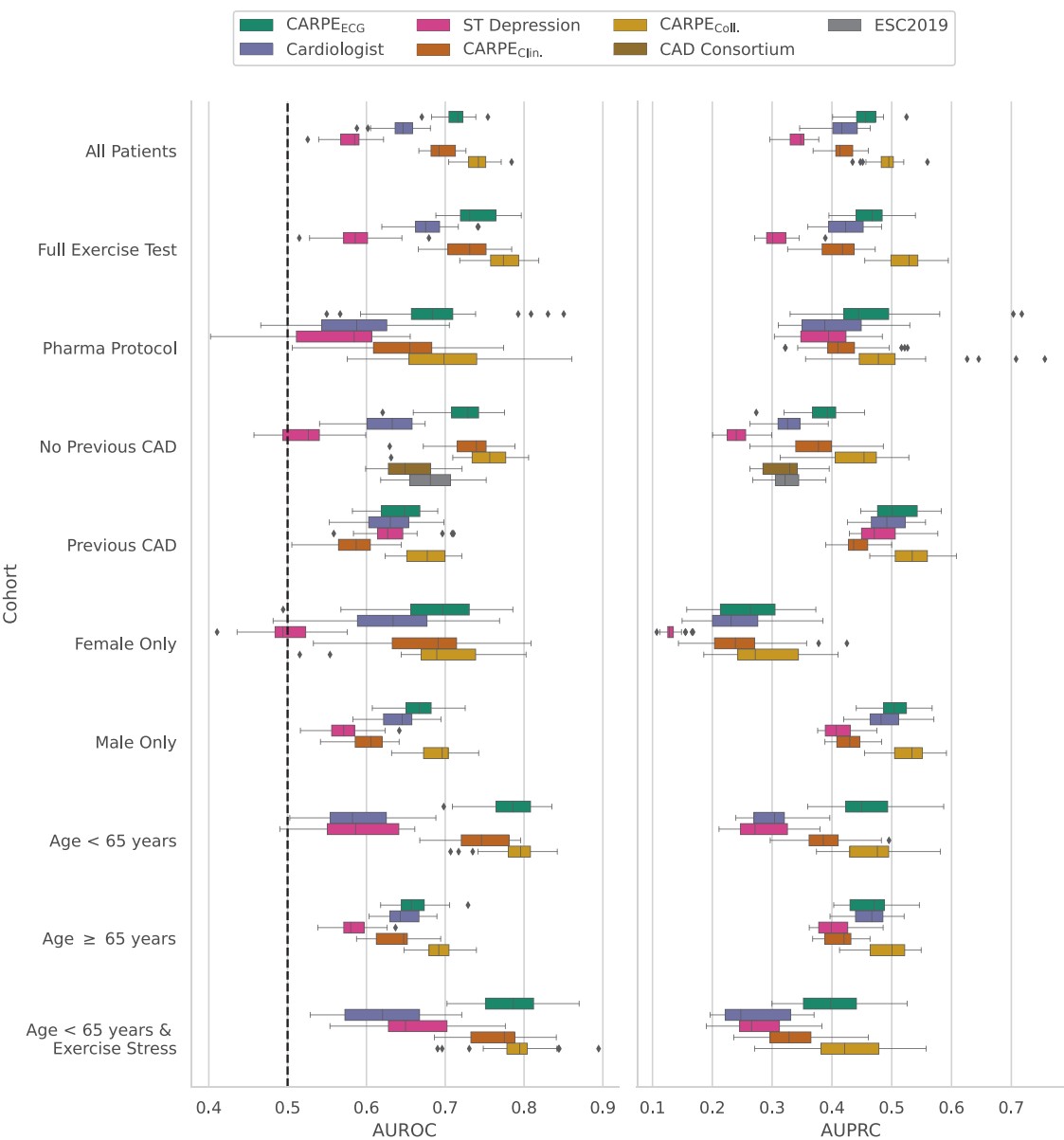

**Fig. 3 | Diagnostic performance subcohort analysis.** Performance breakdown over different subcohorts and $n = 25$ bootstrap draws. The dashed black line indicates the AUROC of a random classifier. Over the full cohort (All Patients), both CARPE$_{Clin.}$ and CARPE$_{ECG}$ reach a statistically significantly higher AUROC than the cardiologist. Additionally, the collaborative approach (CARPE$_{Coll.}$) significantly increases predictive performance over CARPE$_{ECG}$. Please refer to Supplementary Table 6 and Supplementary Fig. 4 for more details. Box plots indicate median (middle line), 25th, and 75th percentile (box). Whiskers extend to points that lie within 1.5 IQRs of the lower and upper quartile. Diamonds are outliers. Error bars in the bar plots indicate 95% confidence intervals. Source data are provided as a Source Data file.

(CARPE$_{Clin.}$)) and not in CARPE$_{ECG}$ ($p = 0.055$, Welch's $t$-test for independent samples, test statistic $= 2.24$). Furthermore, the SHAP distribution of these variables around the value of zero is strikingly different. While CARPE$_{ECG}$ exhibits many values comparatively close to zero (i.e., there are patients for which the respective features have no significant impact on the model's final prediction), both CAD history and "sex" have a large impact on the model's prediction in all patients for the conventional ML model (i.e., the distance to zero for both positive and negative SHAP values is substantial). Additionally, both features show a distinctive separation: each variable instance always leads to either a positive (male and presence of CAD history) or negative (female and absence of CAD history) SHAP value. We observe another distinctively different behaviour in the distribution of SHAP values for the "age" feature. The conventional ML model has learnt an age threshold of 70 years, which, when exceeded, leads to mostly

positive SHAP values (i.e., it contributes to predicting the presence of fCAD) and vice versa. CARPE$_{ECG}$, on the other hand, exhibits a distinctive bell shape around zero, indicating the reduced impact of this variable. While this bias is likely due to the reduced fCAD prevalence of younger patients, the DL model exhibits a more stable behaviour with respect to this variable. The conventional ML model's reliance on young age as a strong indicator of the absence of fCAD turns out to be detrimental when evaluated on external data, which consists of significantly more young patients (see Fig. 5). This underscores the need for explainability and trustworthiness in assessing ML models; if unaddressed, these aspects may preclude clinical applicability.

In addition to performing a population-wide feature relevance analysis, SHAP values allow for sample-specific analyses. In panel b of Fig. 4, we show a case study of an 83 year-old male patient with no previous CAD. We envision that in a future clinical implementation of

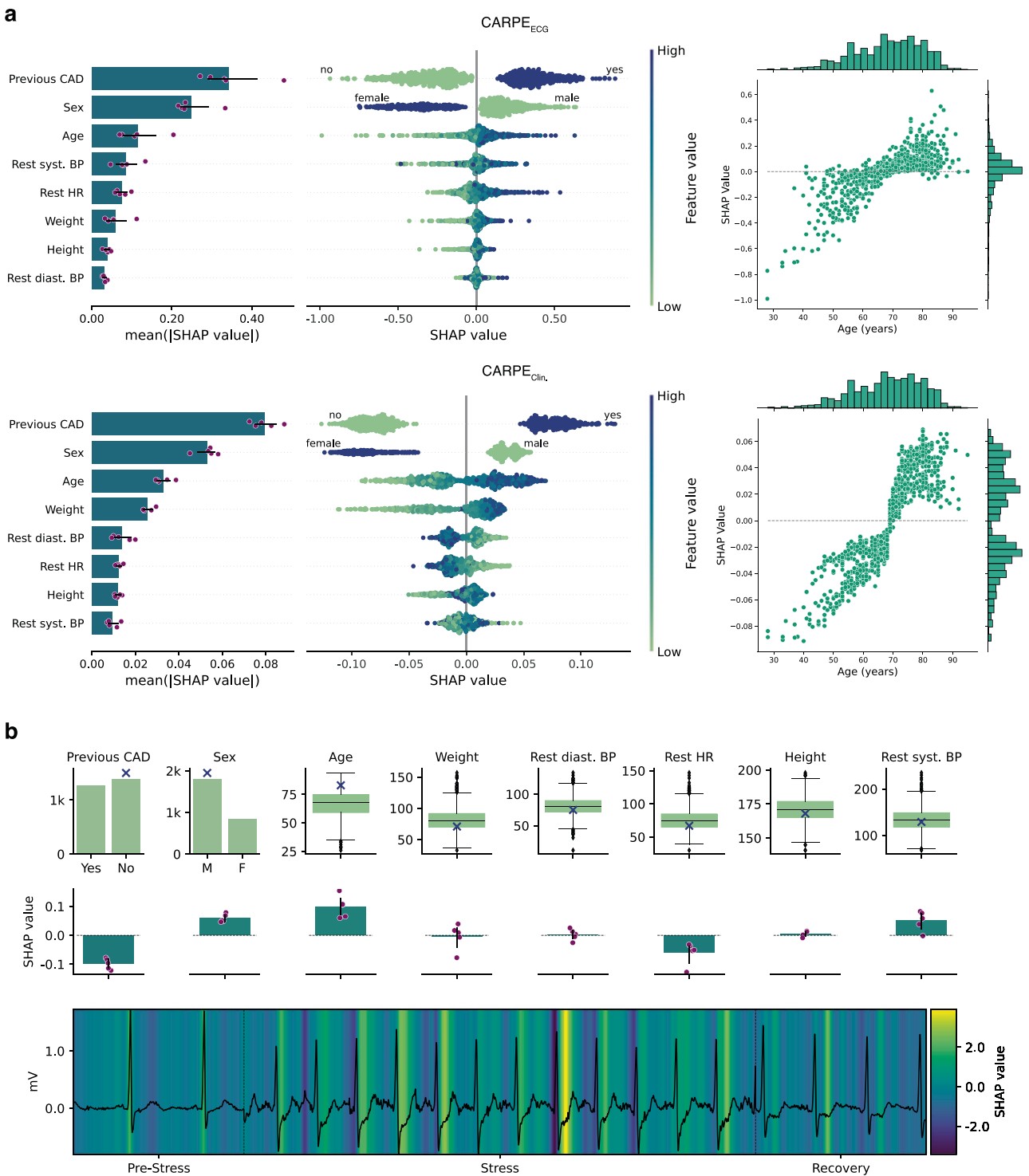

**Fig. 4 | SHAP value analysis. a** Bar plots show the mean absolute SHAP value for all clinical variables used by our predictors. Purple scatter plots show individual data points. CAD history and sex are the most important clinical features for both classifiers. The central scatter plots show the impact individual feature values have on the prediction score. High feature values are depicted in a dark blue, low values in a light green. SHAP values for an existing CAD history are always positive. Similarly, SHAP values of the "sex" feature are always positive for male patients. We depict SHAP value distributions over all ages in the scatter plots on the right-hand side. **b** SHAP values for clinical variables and one 2-6-2 sequence of a patient. The first row shows the feature distribution of the development data set (*n* = 2648) in green. The blue cross marks where in the distribution the patient lies. Second row: SHAP values for the specific patient for each feature over *n* = 5 splits. The absence of

a CAD history and the resting heart rate of 67 BPM result in negative SHAP values. The patient's sex (male), his age, and systolic blood pressure at rest are associated with higher SHAP values. Last row: One of the patient's 2-6-2 sequence (black) with the SHAP values of each individual measurement in the background. We show negative SHAP values in dark purple and positive ones in yellow. Dashed black lines mark the borders of pre-stress, stress, and recovery samples. The largest areas of high SHAP values concentrate in the stress phase around the ST-segment. Error bars in all plots indicate 95% confidence intervals over all models from all five splits. Box plots indicate median (middle line), 25th, and 75th percentile (box). Whiskers extend to points that lie within 1.5 IQRs of the lower and upper quartile. Diamonds are outliers. Bar plots show the mean over *n* = 5 test splits with error bars indicating 95% confidence intervals. Source data are provided as a Source Data file.

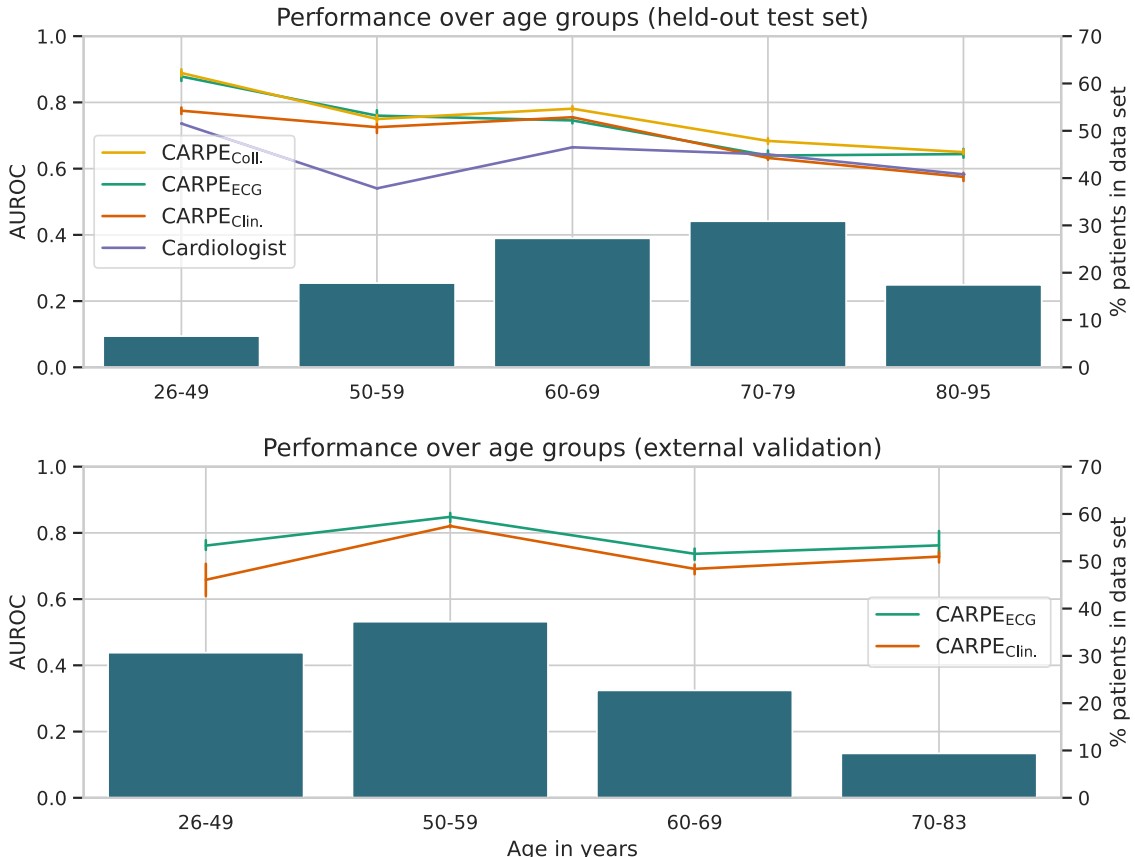

**Fig. 5 | Diagnostic performance over age groups.** On the x-axes, we show different age groups in the held-out test and external validation set. Left y-axes: area under the receiver operator characteristic (AUROC). Error bars indicate 95% confidence intervals around the mean. Right y-axes: percentage of patients who comprise the respective subgroup of the x-axis. No cardiologist's judgement is available in the external validation set, hence CARPE$_{Coll.}$ cannot be evaluated. The

performance difference between random forest and CARPE$_{ECG}$ is strongest in the external validation set due to the conventional ML model relying (too) strongly on the "age" variable. Error bars indicate 95% confidence intervals over all models of all five splits. The number of individuals in each bin are 53, 143, 219, 248, 140 for the held-out test set and 281, 341, 208, 86, respectively. Source data are provided as a Source Data file.

our risk assessment tool, such a dashboard will support the cardiologist to understand better on which basis the model arrived at its prediction (e.g., whether the ECG signal is disturbed or noisy) and the influence of each feature (e.g., SHAP values).

The first row of panel b depicts the distributions of the values of all clinical features from the training population. Blue crosses indicate where the patient lies in that distribution. The centre row shows the distribution of SHAP values over five iterations. Moreover, we show the SHAP values of individual measurements in the background of the input ECG in the last row. The mean risk-score CARPE$_{ECG}$ provides for this patient, who was later diagnosed with the presence of fCAD, is 0.77. We show positive SHAP values in yellow, negative ones in dark purple.

Notwithstanding their opposing signs, among the clinical variables, both the absence of a previous CAD and the patient's age contribute most to the model's prediction (-0.1 and 0.1, respectively). The normal resting heart rate of 67 is associated with a lower risk score (mean SHAP value: 0.07). While weight, height, and diastolic blood pressure influence the model only marginally, the fact that the patient is male contributes most towards a higher risk score. Similarly, the patient's age lies above the upper quartile of the training distribution, pushing the model toward predicting a higher score. Lastly, the systolic blood pressure (129 mmHg) also contributes to the prediction of the positive class. The largest contribution that increases the model's output comes from the ECG. The SHAP values attributed to certain measurements and segments in the ECG might change throughout the different phases of stress testing. In sum, the mean SHAP value for the

whole signal is 2.31. The highest SHAP values can be observed in the part of the input signal that comes from the stress phase of the examination. Measurements around the R-peak during rest and, more strikingly, around the ST-segment in the stress and partially in the recovery phase are associated with higher SHAP values than other segments of the ECG. The latter observation is a data-driven and a priori domain-agnostic confirmation of the relevance of ST-segment depression in the diagnosis of fCAD. This is underlined by the fact that in the pre-stress phase, where almost no ST-segment depression is visible, SHAP values around the ST-segment are close to zero. Conversely, negative SHAP values, in line with conventional medical understanding, are observed in the T-wave region during rest, the PR interval during stress, and prominently at the ventricular activation or R-wave peak time. This case study and the relevance of ST-segment depression for the prediction of higher risk scores is supported by a population-wide SHAP analysis in Supplementary Figs. 7 and 8.

## CARPE$_{ECG}$ generalises to unseen data across countries and modalities

To validate our neural network's generalisation capabilities, we compute its predictive performance on an external validation data set containing 916 consecutive patients referred for exercise myocardial perfusion single photon computed tomography. Referral reasons included non-anginal chest pain, atypical angina, presence of risk factors, or shortness of breath. This data set was retrieved through the THEW data repository[48] (SUI: E-OTH-12-0927-015); it differs from the development data in several key characteristics: First, instead of

recording the stress test ECG using bicycle ergometry, it was captured by a treadmill exercise test. Therefore, the resulting signal is subject to noise from walking movements rather than the cycling activity. Second, with a mean age of 55 years, the population in the external data set is significantly younger ($p = 1.5E\text{-}121$, one-sided Welch's $t$-test, test statistic = 25.39) than the internal study cohort (held-out test set) whose patients are on average 68 years old (see Supplementary Fig. 9 for a complete comparison of all clinical variables). Lastly, the prevalence of ischaemia in the internal cohort is significantly higher compared to the external validation set (7.5%).

As shown in Supplementary Table 7, both approaches reach a good overall diagnostic performance and perform better on the external data set than on the internal held-out test set. CARPE$_{ECG}$ outperforms the conventional ML model in both AUROC ($0.80 \pm 0.01$ vs. $0.75 \pm 0.004$) and AUPRC ($0.28 \pm 0.02$ vs. $0.19 \pm 0.01$). We attribute the higher predictive performance of the DL model to the fact that despite coming from a different modality, ECG signals are not fundamentally different among different populations, making it a robust and reliable input signal.

In Fig. 5, we contrast predictive performance on different age groups in both internal and external validation data. In patients who are younger than 70, both computational approaches consistently outperform the cardiologist in terms of diagnostic accuracy. However, for the stratum that makes up the majority of the data set (ages 70–79), pure computational prediction and human judgement individually perform comparably. However, their combination (CARPE$_{Coll}$) significantly ($p = 8.1e\text{-}4$, one-sided Welch's $t$-test, test statistic = 7.58) increases diagnostic performance over the cardiologist's judgement and over CARPE$_{ECG}$ ($p = 0.001$, test statistic = 4.73). The two extremes of the age distribution exemplify how the random forest's cutoff of 70 years (see SHAP analysis) leads to detrimental performance: The further away a patient group lies from the cutoff, the bigger the performance difference between CARPE$_{ECG}$ and CARPE$_{Clin.}$ becomes. This is even more pronounced in the external validation cohort, where the differences in mean AUROCs (i.e., 10.3 percentage points) are the largest in patients between 26 and 49 years of age.

## Discussion

We derived and validated two ML models for the safe risk-stratification of patients with suspected fCAD. The models were developed using basic clinical information and raw stress test ECG signals. Their performance was compared to a numerical risk estimate by the treating cardiologist after stress testing. Justified by their good predictive performance, we find both models to potentially be useful risk-stratification tools for (a) the primary care setting where cardiac stress testing is not always performed but there is access to the relevant clinical information and (b) the secondary and tertiary setting where stress testing is performed and relevant clinical information is available. Both ensemble learning based on basic clinical information and deep learning based on clinical information and stress ECG signals outperformed the cardiologist's numerical risk assessment after stress testing as well as currently employed risk scores in predicting the presence of fCAD. At a rule-out probability threshold of <15% as used in current clinical practice guidelines[8,21], compared to the cardiologist, the use of the deep learning model enabled a potential average reduction of myocardial perfusion imaging of 15% without increasing the rate of false negatives at 89% sensitivity and 90% NPV (vs. cardiologist 87% sensitivity and 83% NPV). This was partly due to a consistent risk over- and underestimation at the tails by the cardiologist and a lower diagnostic accuracy in comparison (see Supplementary Fig. 6). This shows that calibration is not only a challenge when training ML models but also that the numerical probability estimates from experts must be interpreted with caution. As observed in other studies in ML for healthcare[20,22–24,49], we show that conventional ML models based on clinical data alone can be effective predictors and on par with

deep learning models when considering the whole data set at once. However, in the context of fCAD prediction, we observe that when stratified into clinically relevant patient subgroups or validated externally, deep learning models consistently yield increased diagnostic performance in most strata. This is likely a direct consequence of phrasing the prediction task as a multi-task problem, thereby preventing overfitting, and the additional data source (ECG signal) allowing the network to learn nuances a conventional ML model cannot capture. For instance, our SHAP analysis revealed that compared to the deep learning model, the ensemble model heavily relied on the "sex" and the "age" feature, with the latter rendering it less generalisable in external validation. A post-hoc interpretability study of the neural network confirmed the relevance of ST-segment depression when predicting fCAD and highlighted the usage of feature attribution methods (such as SHAP values) as potential biomarker discovery tools. In line with previous work[50], we therefore recommend that any predictive model in cardiology should be assessed internally and externally in terms of (1) its general predictive performance, (2) its effectiveness in clinically relevant subgroups, (3) the relevance of its features, and, if possible, (4) in the context of a human baseline or common clinical practice. In particular, evaluating the degree to which presenting feature attribution values to clinicians may impact their decision-making will be the subject of future prospective studies. Our study showed that combining both computational approaches with the cardiologist's assessment using logistic regression analysis further increased predictive performance by potentially cancelling out each other's weaknesses, such as algorithmic or cognitive biases. This combined approach led to a mean AUROC increase of four percentage points over the DL model in patients with a CAD history and an increase of 17 percentage points over the cardiologist numerical prediction in patients below the age of 65 who possessed the capacity to undergo the stress test unaided by pharmacological support. Several limitations should be considered when interpreting our findings. Although we have used a stringent methodology to adjudicate the presence or absence of fCAD, we still may have misclassified a small number of patients. Reflecting clinical practice, the expert interpretation of fCAD was not blinded to clinical and stress ECG data, which could have led to an overestimation of these features. Nevertheless, the model's discriminative performance remained consistent across patients, regardless of whether they received invasive coronary artery assessment or not (Supplementary Figs. 2 and 3), indicating a minimal impact of this non-blinded approach. The model was developed in symptomatic patients referred to a tertiary hospital. During study enrolment, MPI-SPECT/CT was the standard non-invasive imaging modality and was applied to patients with a wide range of pre-test probability for CAD. Based on the pre-test probabilities employed in the current ESC guidelines for the diagnosis and management of chronic coronary syndromes[21], 29% of the patients included in the derivation and internal validation cohort would be classified as low risk (<15% probability). As in most other cohorts enroling consecutive patients with suspected CAD, women were underrepresented in the overall cohort. Accordingly, some of the subgroup analyses may have been underpowered in female patients. Similarly, patients of African or Asian descent were underrepresented in this study, and potential differences between these groups cannot be addressed. In the derivation cohort, 26% of patients were below 60, and 7.6% were below the age of 50. Therefore, the results of this study might not apply to very young patients. While the value of more advanced neural network architectures (e.g., attention-based) and ensemble methods (e.g., gradient-boosted trees) may also be explored in the future, prospective clinical studies must be prioritised to establish the clinical value of the CARPE models, interpretability dashboards, and collaborative machine learning.

We integrated the clinical judgement of physicians into our machine learning model using logistic regression, which further

increased its accuracy. However, it is important to acknowledge the limitations of logistic regression analysis and that in real-world clinical practice physicians are unaware of the influence their numerical predictions have on the model's score. The extent to which this knowledge gap influences subsequent risk assessments and thus the model's performance in real-world clinical practice remains an open question. Moreover, clinical utility and the generalisation capabilities of our method are affected by distribution shifts in the input data. In particular, the dependency of clinical judgement on the practitioner's level of experience and their intuitive understanding of the patient's medical history warrants a recalibration of both the ensemble model and the logistic regression before deployment in novel clinical environments. Thus, the observed improvement in performance through logistic regression analysis may not directly reflect the clinical utility or practical applicability of its predictions in healthcare settings. For our collaborative approach to achieve this, careful model recalibration, score interpretation, as well as continuous monitoring of clinical outcomes will be required. As a leading cause of mortality and morbidity worldwide, fCAD is affecting an ever-increasing patient population. With the concurrent demographic ageing in most high- and middle-income countries, there is a major clinical need for safe, accessible, effective, and cost-efficient risk stratification tools to identify patients. Our research underscores the potential clinical utility of ML in reducing potentially unwarranted examinations to support clinicians in providing the best possible care for their patients. Ultimately, maximising predictive performance and clinical acceptance most likely necessitates integrating human judgement with ML predictions in some way.

## Methods

The BASEL VIII study was approved by the local ethics committee (swissethics, BASEC, Ethikkommission Nordwest- und Zentralschweiz) under the number EKBB 100/04 and carried out according to the principles of the Declaration of Helsinki.

### Study population

This analysis is part of a large prospective diagnostic study (NCT01838148, clinicaltrials.gov) designed to advance the early detection of inducible myocardial ischaemia[51,52]. Consecutive adult patients referred to the University Hospital Basel, Switzerland for rest/stress myocardial perfusion single-photon emission tomography/computer tomography (MPI-SPECT/CT) with symptoms possibly due to inducible myocardial ischaemia were enrolled between January 2010 to May 2016. During that period MPI-SPECT/CT was the preferred imaging modality in patients with a wide range of pre-test probabilities for functionally relevant CAD[29,52]. All patients provided written informed consent. Participants did not receive any form of financial compensation or equivalent benefits for their participation in this study. Clinical information, including patient characteristics, medications, symptoms, and prior cardiovascular history were documented by physicians using standardised questionnaires and all medical files available. Based on all clinical information prior to testing, the treating physician recorded a subjective clinical assessment regarding the presence of inducible myocardial ischaemia due to CAD on a visual analogue scale with values between 0% and 100%. Supplementary Table 9 shows demographic and clinical characteristics of patients in development and held-out test set. The sex of the patient was based on the medical files of the University Hospital Basel which represent the Swiss civil status register (i.e., "Geschlecht/Sex" as listed on the passport). We only disaggregated data by sex not by gender, as the latter has not been collected.

### ECG preprocessing and feature extraction

To prepare the 12-lead ECG signals as input to our deep learning approach, we first performed a small number of signal preprocessing steps including downsampling, smoothing, and outlier removal. However, as the choice of preprocessing can affect the ECG's morphology[53], preprocessing parameters were determined by a grid search (see Supplementary Fig. 5). Simultaneously, we evaluated the predictive performance of individuals leads and their combination and selected the best performing combination for evaluation on the held-out test set. Secondly, scalability limitations imposed by the neural network architecture require a significant reduction of the length of the ECG input signal from ~500,000 time points (i.e. 15 min) to 5000. For this, we sample 2 s from the beginning of the examination, 6 s from the last 2 min of the stress phase, and 2 s from the last 3 min of the recovery phase and merge them into a single time series (see panel b in Fig. 1) whose information content is dominated by the stress phase. This time series, which we refer to as the 2-6-2 sequence, was constructed up to twenty times per patient by sampling the subsequences at different time points. Such summary sequences represent a compromise between expressivity (each sequence contains information from the warm-up, stress, and recovery phase) and computational scalability.

### Exercise stress testing protocol and ECG raw data acquisition

Resting heart rate, blood pressure, and 12-lead resting ECG were recorded before exercise. A standardised, stepwise, and symptom-limited upright bicycle exercise test was performed[54,55]. Beta-blockers and antianginal medication were paused for at least 48 h and nitrates for at least 24 h before testing. Exercise stress testing was considered conclusive if 85% of the age predicted maximum heart rate was reached. If this was not feasible, physical exercise was stopped and patients were switched to an adenosine or dobutamine pharmacologic stress testing protocol[51,55,56]. In patients in whom physical stress testing was contraindicated or the target exercise performance was not reached, pharmacological testing was performed. After testing and blinded to the MPI-SPECT/CT results, the treating physician once more recorded a clinical post-test probability regarding the presence of fCAD on a visual analogue scale (0–100%). The 12-lead ECG signals were recorded with two different devices (Schiller AT-110 and Schiller CS-200 Excellence) at 500 Hz and 1000 Hz with a minimal resolution of 5 µV/bit and a minimal diagnostic signal bandwidth of 0.05 Hz to 150 Hz.

### Adjudication of fCAD

Adjudication of fCAD was based on expert interpretation of MPI-SPECT/CT images combined with information obtained from invasive coronary angiography and fractional flow reserve measurements, whenever available. All patients underwent a routine standard rest/stress dual isotope ($^{201}$Tl for rest, $^{99m}$Tc sestamibi for stress) or a single isotope ($^{99m}$Tc sestamibi for stress and rest) MPI-SPECT/CT protocol. MPI-SPECT/CT images were scored semi-quantitatively using a 17-segment model with a 5-point scale (0 = normal, 1 = mildly reduced tracer uptake, 2 = moderately reduced uptake, 3 = severely reduced uptake and 4 = no uptake). Summed stress score and summed rest score were calculated by adding the scores of the 17 segments in the stress and rest images. Summed difference score was the difference between summed stress score and summed rest score. A summed difference score of at least 2 or positive transient ischaemic dilation ratio (≥1.22 for the dual isotope protocol and 1.12 for the single isotope protocol) was considered as fCAD[27–29]. Summed stress score and summed rest score were derived by visual assessment of two expert readers and compared with the software result. Differences in the visual assessment were resolved by finding consensus. In case of equivocal findings from MPI-SPECT/CT and coronary angiography, an adjudication committee of two independent cardiologists (one interventional cardiologist, one general cardiologist) that were blinded to study biomarker results reviewed the case using all clinically available data. A positive perfusion scan was overruled when coronary

angiography showed normal coronary arteries, while a negative perfusion scan was overruled if coronary angiography (within 3 months) either revealed a high-grade coronary lesion (>75%) or if there was fractional flow reserve below 0.80. In total the adjudication committee reviewed 147 cases or 21% of the 701 patients that underwent coronary angiography within 90 days.

## Neural network architecture

Supplementary Fig. 10 provides an overview of the multi-task learning neural network architecture. While the patient's static data $X_{clin}$ is embedded by a neural network, the ECG data ($X_{ecg}$) is fed into a residual neural network akin to the one used by Ribeiro et al.[15]. The concatenation of the output of the embedding layer for the clinical variables and the output of the residual network serves as input to four subnetworks, each of which is responsible for the prediction of one of the four tasks. Each task has its own loss function. More details about the exact layer definitions can be found in Supplementary Fig. 3.

## ST-segment depression

The human baseline is complemented by an automated determination of ST-segment depression. While this morphological feature is commonly linked to ischaemia[39,40], the exact time points in the ECG at which ST-amplitude is measured varies[4]. We compute ST-segment depression as follows. First, we perform a QRS-delineation using the neurokit2 software package[4,57] on the complete stress test ECG. Then, we determine the mean isoelectric line for each stress phase of a given 2-6-2 sequence. For this, we take the mean of the last $l_{PR}$ milliseconds preceding the Q-wave over all heartbeats in a specific stress phase. Similarly, we determine the mean ST-amplitude for each 2-6-2 stress phase by using the ECG measurement 60 ms after the J-point. The mean ST-segment depression (difference between mean isoelectric line and ST-amplitude) is determined for each stress phase ($ST_{Pre}$, $ST_{Stress}$, $ST_{Rec}$). The differences between $ST_{Stress}$/$ST_{Rec}$ and baseline ST-depression ($ST_{Pre}$) are then aggregated over all 2-6-2 sequences of a patient by using either their mean, median, minimum, or maximum. Importantly, the physiological response to stress may differ among the patients subject to different stress types. Therefore, the parameter grid shown in Supplementary Table 10 is evaluated separately for all three cohorts and all leads.

## Data splits and bootstrapping

We split the data set 3:1 into a development and held-out test set containing 2648 and 874 patients, respectively (see Supplementary Fig. 11). During the development of the model, we had no access to the held-out test set. Access was provided once we fixed all model parameters. The development set was further divided into 5 stratified splits of training, validation, and calibration set, where the latter makes up 10% of the training set. The ratio of training to validation set size is 4:1. Each of five splits of the development set contained on average 36977, 9254, and 5129 train, validation, and calibration 2-6-2 sequences from 1882, 471, and 260 patients, respectively. If not specified otherwise, bootstrapping has been performed to obtain distributions for statistical testing. This was done by pooling all predictions from all five splits and sampling 80% in 25 different draws. Since the cardiologist only scores each patient once, we sampled the same patients for the cardiologist that have been selected for the computational methods in each draw. This way each draw contains predictions for the same patients from the different predictors to be compared.

## Statistics and reproducibility

All p-values for the comparison of performance metrics (i.e., AUROC, AUPRC) were computed on the distributions obtained by the bootstrapping procedure described in the previous section. For this, a one-sided Kolmogorov-Smirnov test was used. Multiple hypotheses are corrected for using Bonferroni correction. When comparing the age

distribution of our data set with the external data set, a one-sided Welch's t-test was used. To compare the odds of obtaining a positive fCAD label with/without a history of CAD, we used a two-sided Fisher's exact test. The comparison of SHAP values was performed over the distributions of five data splits as described above; we used Welch's t-test for independent samples. The development set was split into training/validation/calibration in a stratified manner ($n = 5$), meaning the fCAD prevalence remains the same in all splits. From the original data set, 697 patients were excluded because no digital ECG data was available (see Supplementary Fig. 11). Reflecting clinical practice, the expert interpretation of fCAD was not blinded to clinical and stress ECG data. However, the treating physician was blinded to the MPI-SPECT/CT results when submitting the post-test probability score. With access to the THEW data set (SUI: E-OTH-12-0927-015), all results pertaining to this data set can be reproduced using the publicly available code at https://github.com/BorgwardtLab/CARPE[58].

## Preprocessing, lead selection, and auxiliary task regularisation

All 1000 Hz signals were downsampled to 500 Hz. ECG signals from exercise stress testing are subject to high noise levels from various sources. To assess the influence of noise on classification performance, we consider the following preprocessing schemes: 1. No preprocessing, 2. minimal preprocessing with a high-pass Butterworth filter of order five, and a cutoff frequency of 0.5 Hz followed by moving average smoothing, and 3. a thorough preprocessing pipeline consisting of a wider bandpass filter (0.05 Hz–150 Hz), moving-median subtraction to remove baseline wandering, a Savitzky–Golay filter[59] for smoothing, and winsorizing to deal with spurious outliers.

To evaluate the impact that individual ECG leads, preprocessing, and auxiliary tasks have on predictive performance, we proceeded as follows: First, we used the first development split to determine the most promising leads (in terms of area under the precision-recall curve (AUPRC) on the validation set) by performing a grid search over (a) three preprocessing schemes described above, and (b) learning rate parameters $\eta \in \{0.01, 0.001, 0.001\}$ for all twelve leads individually and in combination. More specifically, we trained $13 \times 3 \times 3 = 117$ neural networks to determine the three best performing leads. The first number accounts for the 12 individual ECG leads plus one configuration that combines all leads. The second number represents three preprocessing schemes and is followed by the number of learning rates that were analysed. Subsequently, we picked the three best-performing leads and their respective preprocessing/learning rate combination to assess the impact of all auxiliary tasks. In order to do so, the performance on the validation set was averaged over all splits on a $5 \times 5 \times 5 \times 3$ parameter grid as shown in Supplementary Table 4. Finally, the best-performing model was enriched with clinical variables to receive the final model, which we evaluated on the held-out test set. The results of this analysis on the validation set of the development data set are shown in Supplementary Fig. 5 and Supplementary Table 5.

## Calibration

Supplementary Fig. 6 depicts the calibration of $CARPE_{ECG}$, $CARPE_{Clin.}$, and cardiologist on both training and held-out test data. The red dashed lines indicate the two decision cutoffs (5% and 15%) as advocated in European and US-American guidelines[8,21]. On the training set, $CARPE_{ECG}$ is almost perfectly calibrated at 5% but slightly overestimates fCAD probability at 15%. On the held-out test set, $CARPE_{ECG}$ remains close to the diagonal but now underestimates the presence of fCAD. The cardiologist underestimates the presence of fCAD at both thresholds and in both data sets, yet performs similarly to $CARPE_{ECG}$ at the 15% on the held-out test set. The ensemble method significantly overestimates the presence of fCAD around the relevant decision thresholds on the training set and reaches best calibration on the held-out test set at the 5% cutoff. At the 15% threshold, however, it continues to overestimate the presence of fCAD. Both computational methods

exhibit a significantly lower Brier score (hence are better calibrated than the cardiologist) on the held-out test set. More precisely, the cardiologist reaches a score of $0.23 \pm 0.009$, the random forest $0.22 \pm 0.004$ ($p = 3.59E\text{-}5$, one-sided $t$-test, df = 24, test statistic = -4.78), and CARPE$_{ECG}$ $0.18 \pm 0.006$ ($p = 7.97E\text{-}16$, one-sided $t$-test, df = 24, test statistic = -18.15).

### External validation data

The external dataset consists of 927 consecutive patients referred for exercise myocardial perfusion single photon emission computed tomography (SPECT) at Assuta Medical Center and Sheba Medical Center and is a subset of the data presented by Sharir et al.[49] Throughout the baseline, exercise, and recovery phases, a high-resolution 12-lead ECG was continuously recorded using the HyperQ Stress System from BSP Ltd, Tel Aviv, Israel, at a rate of 1000 samples/second with 16-bit resolution and an analogue frequency response of 0.05 to 125 Hz (measurement sensitivity <0.15 V). Patients with a cardiac pacemaker, atrial fibrillation at the time of testing, or a QRS duration equal to or greater than 120 milliseconds were excluded from the study. The exercise procedure was carried out as follows: beta blockers and calcium channel blockers were discontinued at least 48 h prior to the test, and a symptom-limited treadmill exercise test was conducted using the Bruce protocol.

In contrast to our internal training and held-out test set, the external data set is not annotated with the stress phases (pre, stress, recovery). We therefore compute the heart rate for each minute of the ECG signal and use its maximum as the end of the stress phase. This allows us to extract 2-6-2 sequences that are equivalent to the training data. Additionally, we exclude patients for which the required clinical variables are missing. If the value for a single variable is missing, we exclude the patient. This is the case for the ground truth label (missing in nine patients), the weight variable (missing in one patient), and the height variable (missing in one patient). Supplementary Fig. 9 depicts the distribution of relevant clinical variables of the internal and external data sets. The biggest difference between both data sets can be observed in the age variables (patients from the internal data set are significantly older) and the fCAD/ischaemia prevalence.

### Additional SHAP analysis

To measure the impact ECG segments have on the prediction, we summarize their SHAP values by summing them to express a segment's contribution in a single scalar value. We perform this aggregation for all ECG segments in all 2-6-2 sequences. We then stratify each segment by its stress phase to investigate whether the origin of the segment (in terms of the stress test phase) influences the attributed contribution. Supplementary Fig. 7 shows the result of this analysis in patients for which a low and high CAD probability was predicted. Overall, we observe more segments with SHAP values deviating from zero in the higher-risk population. In particular, the ST-Segment in the stress phase is associated with particularly high SHAP values. For the population for which a low risk was predicted, the SHAP values from the QRS complex from the stress phase contribute, on average, most to the prediction signal. We summarize both the QRS complex and the ST-segment from the respective cohorts in Supplementary Fig. 8.

To investigate whether specific ECG patterns exist that contribute to a lower/higher predicted CAD score, we perform the following analysis: If a stress test phase (i.e., Pre/Stress/Recovery) of a 2-6-2 sequence contains an ECG pattern (e.g., P-wave, ST-segment) whose summed SHAP score exceeds/is lower than a threshold, we extract all ECG waves (i.e., from P-wave onset to T-wave offset) from this phase. Each extracted ECG wave contains a representative of the pattern that has a high influence on the model's prediction. Motivated by the results visualized in Supplementary Fig. 7, we choose ECG waves whose QRS-complex has a SHAP score of maximal -0.25 to highlight a pattern contributing to the prediction of the absence of CAD. Similarly, we selected ECG waves whose ST-segment has a SHAP score of at least 0.25 as a pattern that contributes to the prediction of the presence of CAD. Furthermore, we limit this analysis to samples with higher predictive score and apply the same probability thresholds (15% and 85%) as in the previous analysis. We align all waves using dynamic time warping and visualize both the aligned waves (grey) and their mean wave (red) in Supplementary Fig. 8.

### Calibration and performance for patients with and without ICA

Supplementary Figs. 2 and 3 show calibration and predictive performance (AUROC and AUPRC) for patients who did and did not undergo ICA within 90 days. The respective prevalence of fCAD in the held-out test set was 84% and 18%, respectively. In the latter cohort, the shown predictors are calibrated similarly to the full cohort (viz. Supplementary Fig. 6). In the subgroup of patients who underwent ICA within 90 days (i.e., overall high-risk patients), we observe that all predictors, including the cardiologist, underestimate the presence of fCAD. Considering AUROC, CARPE$_{ECG}$ exhibits a better mean predictive performance in both cohorts. In patients with ICA, the mean performance gain is 18 percentage points (CARPE$_{ECG}$: $0.72 \pm 0.06$, Cardiologist: $0.54 \pm 0.06$) and 11 percentage points (CARPE$_{ECG}$: $0.72 \pm 0.02$, Cardiologist: $0.61 \pm 0.03$). Furthermore, in patients who did not undergo ICA within 90 days, the combination of cardiologist and deep learning method, CARPE$_{Coll.}$ modestly increases mean predictive performance from 0.72 to 0.74.

### Statistical interaction tests by subgroups

Supplementary Fig. 4 provides a statistical interaction analysis based on the performance increase of CARPE$_{ECG}$ over the cardiologist. On the complete held-out test cohort, this increase is, on average, 8 percentage points. The only variable showing a statistically significant ($p = 0.0067$, two-sided $Z$-test) effect is "Age," where the performance difference between CARPE$_{ECG}$ and the cardiologist is significantly higher in patients <65 years of age compared to patients who are at least 65 years old.

### Data collection and data analysis software

The 12-lead ECG signals were recorded with two ECG machines, namely a Schiller AT-110 and Schiller CS-200 Excellence. Exported.ful and.xml files were analysed with the standard xml module of python 3.8 and numpy at version 1.24.4. ECG preprocessing was performed using the 'signal' module of scipy at version 1.10.1. QRS Delineation was performed using the matlab ecg-kit version 1.4.0.0. To train our models and analyse and visualize the data, we used the following python libraries and versions ipython version 8.12.3, jupyter version 1.0.0, matplotlib version 3.7.3, networkx version 2.8.8, notebook version 7.0.4, pandas version 2.0.3, pytorch-lightning version 1.2.1, scikit-learn version 1.2.2, scipy version 1.10.1 seaborn version 0.13.0, pytorch version 1.6.0, torchmtl version 0.1.9.

### Reporting summary

Further information on research design is available in the Nature Portfolio Reporting Summary linked to this article.

## Data availability

The data that support some of the findings of this study are not openly available due to reasons of sensitivity of patient data and are available from the corresponding author (christian.mueller@usb.ch) upon request. The request should include the name and full contact information of the person and institution requesting the data, the specific identification of the data being requested and the purpose of requesting the data. Data requests under agreement will be considered for purposes of reproducing the data presented herein, subject to appropriate confidentiality obligations and restrictions. The timeframe for response to requests is estimated to be four to 8 weeks and

restrictions imposed on data use will be individualized by case-by-case data use agreements. The data resides in the secured IT infrastructure of the University Hospital Basel and respective files can be shared after anonymization upon individual request. Data used for external validation was provided by the Telemetric and Holter ECG Warehouse of the University of Rochester (THEW), NY. It cannot be made public by the authors. To obtain access, interested parties must register with the THEW project (http://thew-project.org/), submit a research proposal, and fill out the data usage agreement for the dataset with identifier E-OTH-12-0927-015. For-profit organisations may also purchase the data set for an access fee as detailed on the website. The authors declare that all data supporting the findings of this study which are not protected by patient privacy considerations, are available within the paper, its supplementary information files and downloadable files deposited at figshare (https://doi.org/10.6084/m9.figshare.25514644).

## Code availability

Preprocessing scripts, trained neural network model checkpoints and random forest classifier are publicly available at https://github.com/BorgwardtLab/CARPE[58].

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

## Acknowledgements
This study was supported by the Alfried Krupp Prize for Young University Teachers of the Alfried Krupp von Bohlen und Halbach-Stiftung (K.B.). The founders had no influence on the study question or design in any way. Data used for this research was provided by the Telemetric and Holter ECG Warehouse of the University of Rochester (THEW), NY.

## Author contributions
C.B., J.E.W., B.R., I.St., K.B., C.M. designed the experiments; J.E.W, I.St., K.R., I.Sc., M.J.Z., C.M. collected and provided both clinical and ECG data; C.B. and I.St. preprocessed the raw ECG signals with contributions from B.R. and J.E.W.; C.B. and B.R. developed and implemented the machine learning pipelines; C.B. performed all experiments with contributions from B.R., J.E.W., I.St., K.B., C.M.; J.E.W. and C.M. performed the clinical interpretation of results and provided C.B., B.R., K.B. with relevant clinical context. C.B. created all figures with support from B.R., J.E.W., K.B.; C.B., B.R., C.B. and J.E.W. performed statistical analyses with contributions from K.B.; K.B., C.M., J.E.W. conceived and directed the project; C.B., J.E.W., B.R., C.M., K.B. wrote the manuscript with the assistance of feedback of all the other co-authors.

## Funding

## Competing interests
J.E.W. has no conflict of interest to declare regarding this project and reports grants from Swiss Heart Foundation (FF19097 and F18111) and from the Swiss Academy Medical Sciences. C.M. has no conflict of interest to declare regarding this project and received research support from the Swiss National Science Foundation, the Swiss Heart Foundation, the KTI, the University of Basel, the University Hospital Basel, Abbott, Beckman Coulter, Brahms, Idorsia, Novartis, Ortho Clinical Diagnostics, Quidel, Roche, Siemens, Singulex, and Sphingotec as well as speaker honoraria/consulting honoraria from Amgen, AstraZeneca, Bayer, Beckman Coulter, Boehringer Ingelheim, BMS, Idorsia, Novartis, Osler, Roche, Sanofi, Siemens, and Singulex. The other authors declare no competing interests.
