## [Peer Review File · Nature Communications]

REVIEWER COMMENTS

Reviewer #1 (Remarks to the Author):

In the study, “Enhancing the diagnosis of functionally relevant coronary artery disease with machine learning”, Bock and colleagues present a multitask model to improve diagnostic decision-making using clinical profiles and stress-testing ECGs. There is substantial clinical utility in defining ECG parameters that allow better risk stratification, especially where more advanced imaging is limited. However, I have some comments about the approach's success in achieving the goals:

- 1) The authors report an AUROC of 0.71 for their best-performing model to risk stratify. At face value, this is a modest performance. While better than clinicians, the NPV of 0.89, observed for the 15% threshold, will be challenging for clinical practice, especially when this approach aims to reduce unnecessary testing. The authors should more clearly justify the optimal thresholds needed for such a validation procedure.
- 2) The data sources in both internal and external validation data require more details. The internal validation data are part of a single hospital cohort study – what were the various clinical settings where the patients presented? The external validation data paragraph does not define any of the aspects clearly – were these patients consecutive? What was the care setting? The CAD history was assumed to be negative – why was that? Why were patients dropped for 1 missing observation? How many patients were dropped this way? Several aspects need substantially better calibration.
- 3) There is some value in traditionally low-risk groups – those that can perform exercise stress tests (as opposed to pharmacological) and younger age. What were the full model performance characteristics in these populations? This is pertinent as the modestly higher AUROC in these groups should be further supported by data on clinical meaningfulness.
- 4) The model identifies common predictors often used in subjective risk stratification and decision-making (prior CAD, age, sex, and other similar features) – this is helpful. However, the ECG features in SHAP analysis are not adequately addressed. Are there aspects from the SHAP synthesis that identify ECG components of interest? The example provided, and the associated discussion is unclear—are there at least positive and negative examples that are meaningful, which could be used to strengthen the synthesis? Also are there summary statistics that can be presented across all positive and negative studies to make the assessment more objective?
- 5) The outcome definitions require further details in the study. The authors said a stress imaging/angiography-based assessment was made by the cardiologist. But since the decisions on further testing were clinically indicated, it is important to know what proportion of the population had the full cascade of testing. And what were the findings of tests.
- 6) Several risk thresholds are presented, but the underlying approaches' reliability information is limited. Since the authors frequently refer to risk stratification before even evaluating the model findings, the reliability of the approach to pre-test stratification of risk is important.

7) The authors state (line 154): “Finally, we combine predictions from both the statistical model and deep learning approach with the cardiologist’s post-test judgment by training a new logistic regression model on all three scores from the training set. We refer to this collaborative approach”. The collaborative model, touted as being the most successful, is unclear to the reviewer. How is this post-test judgment quantified? How is it combined numerically?

8) The 2-6-2 sequence of the stress ECG was constructed up to twenty times per patient for training the model. The authors should clarify the influence of such an approach on application to a real-world setting. How was this applied in the validation setting?

9) The paper is somewhat dense, and many aspects of the modeling and decisions are not well explained. This leads to a challenge in understanding the value of some of the findings when they are presented.

10) The tables and figures are similarly not adequately positioned, and the reviewer was challenged in finding the content essential to interpreting the study. If the set-up of the study and the order of presentation was better presented, I believe the readership will benefit substantially.

Reviewer #2 (Remarks to the Author):

This paper is addressing an early detection technique for Functionally relevant coronary artery disease (fCAD) that can result in premature death or nonfatal acute myocardial infarction. Two machine learning (ML) techniques were used to predict fCAD: initial one is using a small number of static clinical data, whereas the later one leverages electrocardiogram (ECG) signals from exercise stress testing. The results demonstrate that ML can significantly outperform cardiologists in predicting the presence of stress-induced fCAD in terms of area under the receiver operating characteristics.

The paper is well written and provide proper justification of the methods used. The dataset (3522 patients for ECG) is very important feature of this study and it is properly validated by cardiologist.

Machine Learning techniques used in the paper are conventional ML techniques that are used in different applications.

Overall, paper is good and provide good framework how to complement cardiologist by using this framework.

My only criticism with the results presents for younger age or less than 60 years of age population. As the dataset’ median value of age is 66.7 years with standard deviation is 11.1 that how you can predict model for lesser age.

In Fig. 2, Sensitivity and Precision were shown, It is good idea to present the Specificity, Recall and Accuracy and F1 score will presented as well.

Overall it is good paper.

Reviewer #3 (Remarks to the Author):

SUMMARY

Bock et al. present the results of a retrospective analysis evaluating the predictive performance of machine learning for detection of functionally significant coronary artery disease. The authors demonstrate that machine learning can outperform cardiologist estimations of the risk of functionally significant CAD, potentially decreasing the number of patients that would require further risk stratification. Additionally, there seemed to be benefit from combining ML predictions with cardiologist estimates.

GENERAL COMMENTS FOR THE AUTHORS

The authors have performed a comprehensive study which includes model development and external testing for predicting 'functionally relevant' coronary artery disease. This could potentially help physicians choose between functional or anatomic imaging approaches. The most novel aspect of the model is that it incorporates stress ECG data automatically, which can be challenging clinically given the high frequency of noisy baseline data. However, the importance of this is minimized from a clinical perspective because the stress ECG is typically collected during the functional imaging test (instead of performing an exercise stress prior to deciding on which imaging test to pursue). The model more accurately predicts the presence of functionally relevant CAD compared to expert clinical judgement with stress ECG results. There are a few similar studies which have been performed with machine learning (Most similar study, DOI: 10.1007/s12350-022-03012-6; internal testing n=20,418 from 5 sites; external testing n=9,019 from two sites; AUC 0.76 in external testing); (doi: 10.1155/2021/3551756; n=2503); (<https://link.springer.com/article/10.1007/s12350-018-1304-x>); (doi: 10.2196/16975); (DOI: 10.1016/j.cvdhj.2022.02.002). The first study has larger and more diverse training and external testing populations. The authors present another finding of significant interest; integrating expert judgement with the ML prediction improved performance (although there are issues with the current methodology as outlined in point 3 below).

A major concern is that the primary outcome, 'functionally relevant' CAD is a combination of two outcomes (myocardial ischemia by SPECT or obstructive CAD on cath) without clear delineation of how many patients were classified with each method. Selective referral to invasive angiography induces some bias in the study since some clinical aspects which may be utilized by the ML model (ECG findings, symptoms) also influence decisions to pursue invasive angiography. Notably, the previously mentioned studies did not include hybrid outcomes such as the one used by Bock et al.

SPECIFIC COMMENTS FOR THE AUTHORS

- 1) The model is developed in a cohort of patients who were referred for SPECT MPI (which is typically intermediate or higher risk). This is not the same population as the screening population that the authors are discussing in the Introduction, limiting generalizability of the model.
- 2) The definition of "functionally relevant" coronary artery disease outlined by the authors in the Introduction (symptoms of ischemia, death or MI) is not the same as the definition used in the study.
- 3) The cardiologist judgement was combined with the ML prediction using a logistic regression model.

Did the authors consider including it as a feature for the ML model instead? Additionally, why not provide the cardiologists with the results to see how this would change their post-test probability estimate (particularly relevant since this is how it would most likely be implemented)?

4) The authors utilized cardiologist judgement using a visual scale after the EST. What factors were the physicians using? Did any perform formal calculations of pre and post-test probability? Do you have any sense of inter-rater reproducibility for these estimates?

5) There are several available risk scores that could be used to predict the likelihood of obstructive CAD (updated Diamond Forrester, CAD consortium scores, etc.). Consider including comparisons to these scores as supplemental results.

MINOR POINTS:

6) What proportion of patients underwent cath to adjudicate functionally relevant CAD? How many cases were “equivocal” on cath?

7) In the “Data collection, label generation, and robustness”, should state that the external population is also a SPECT MPI referral cohort.

8) The introduction should provide some mention of coronary artery calcium scoring or CCTA.

9) Consider explicitly stating the central adjudication of functionally significant CAD was done blinded to demographics and ECG stress data (assuming this is the case), if not discuss potential bias related to this.

10) Consider removing unnecessary abbreviations (w.r.t. ; l.h.s. etc.)

11) It does not seem appropriate to ‘upsample’ patients with functionally relevant CAD in the external testing population (Page 15). This defeats the purpose of external testing (to see if the model works in other populations) by trying to make the populations more similar. Additionally, since you upsampled this population, it is misleading to say that the prevalence was 6% in Tables.

12) What was the definition of positive TID ratio?

13) Consider including a comparison to the previous work discussed above

Reviewer #1 (Remarks to the Author):

In the study, “Enhancing the diagnosis of functionally relevant coronary artery disease with machine learning”, Bock and colleagues present a multitask model to improve diagnostic decision-making using clinical profiles and stress-testing ECGs. There is substantial clinical utility in defining ECG parameters that allow better risk stratification, especially where more advanced imaging is limited. However, I have some comments about the approach's success in achieving the goals:

1) The authors report an AUROC of 0.71 for their best-performing model to risk stratify. At face value, this is a modest performance. While better than clinicians, the NPV of 0.89, observed for the 15% threshold, will be challenging for clinical practice, especially when this approach aims to reduce unnecessary testing. The authors should more clearly justify the optimal thresholds needed for such a validation procedure.

R: As suggested by Reviewer 1, we have clarified the presentation of the various thresholds. Currently, no "optimal" threshold is defined in the literature, so we chose to present performance across a broad range of potentially clinically relevant thresholds. The current ESC guidelines for the diagnosis and management of chronic coronary artery disease classify patients with a probability greater than 15% (but less than 65%) as the group most likely to benefit from non-invasive testing. For patients with a CAD probability between 5% and 15%, testing may be considered by the clinician after assessing the overall clinical likelihood. **Changes made:** p. 7.

2) The data sources in both internal and external validation data require more details. The internal validation data are part of a single hospital cohort study – what were the various clinical settings where the patients presented? The external validation data paragraph does not define any of the aspects clearly – were these patients consecutive? What was the care setting? The CAD history was assumed to be negative – why was that? Why were patients dropped for one missing observation? How many patients were dropped this way? Several aspects need substantially better calibration.

R: As suggested by Reviewer 1, we added further details concerning the validation cohorts. **Changes made:** Internal validation cohort: p. 3, External validation cohort: pp. 4, 32.

We agree with the reviewer's comment that making the assumption of a negative CAD history requires justification. Initially, we were not able to identify the respective variable in the external data set. After further investigation, however, we confirmed that a “CAD history” variable is present in the external data set, and we reran our external validation experiments. Additionally, we

revisited our initial data cleaning steps and dropped an unnecessary requirement to only include patients for which a conclusive ECG result was present, thereby dropping patients with “inconclusive” and “non-diagnostic” results. As this variable only pertains to the ECG analysis, there is no need to drop them. After removing this requirement, nine patients were dropped because they had no ground truth label, one patient was dropped because of a missing height measurement, and one patient was dropped because of a missing weight measurement. In total, we dropped 11 out of 927 patients. After the revisited preprocessing and the inclusion of the CAD history, **both our methods gained in performance**. On the original (i.e., not upsampled) data set, CARPE_{Clin.} reaches a mean AUROC of 0.75 (before 0.73) and AUPRC of 0.19 (before 0.15). CARPE_{ECG} now achieves a mean AUROC of 0.80 (before 0.77) and AUPRC of 0.28 (before 0.20). In light of this revision, we updated Figure 5 with no notable difference, as well as Extended Data Table 3 and Extended Data Figure 5. In the latter, we would like to highlight that the sex distribution of the external data set is now much closer to the internal data set, which may contribute to the performance increase of both our methods. We thank the reviewer for pointing out these shortcomings. We believe that the additional analysis serves to strengthen our results. Lastly, we expanded the “External Validation Data” section in the Methods and added that the external validation set is a consecutive SPECT MPI referral cohort in the section “Data collection, label generation, and robustness.”

3) There is some value in traditionally low-risk groups – those that can perform exercise stress tests (as opposed to pharmacological) and younger age. What were the full model performance characteristics in these populations? This is pertinent as the modestly higher AUROC in these groups should be further supported by data on clinical meaningfulness.

R: As suggested by Reviewer 1, we added the full model performance in these populations. **Changes made:** Extended Data Tables 2 and 4 as well as Figure 3.

4) The model identifies common predictors often used in subjective risk stratification and decision-making (prior CAD, age, sex, and other similar features) – this is helpful. However, the ECG features in SHAP analysis are not adequately addressed. Are there aspects from the SHAP synthesis that identify ECG components of interest? The example provided, and the associated discussion is unclear—are there at least positive and negative examples that are meaningful, which could be used to strengthen the synthesis? Also are there summary statistics that can be presented across all positive and negative studies to make the assessment more objective?

R: We thank the reviewer for suggesting strengthening our SHAP analysis. In addition to the case study, we performed a cohort-wide (i.e., on the held-out test set) SHAP analyses summarized in Extended Data Figures 6 and 7. In a first

analysis, we investigated the SHAP contribution of individual ECG segments and stress test phases, highlighting the importance of the “stress phase” and its respective QRS-complex for low prediction scores and ST-segments for high prediction scores. We then extracted ECG segments with particularly low and high SHAP contributions and computed their average. Extended Data Figure 7 illustrates a pronounced ST-segment depression that contributes to a high prediction score.

5) The outcome definitions require further details in the study. The authors said a stress imaging/angiography-based assessment was made by the cardiologist. But since the decisions on further testing were clinically indicated, it is important to know what proportion of the population had the full cascade of testing. And what were the findings of tests.

R: As suggested by Reviewer 1, we added further information concerning the outcome definition and use of angiography-based assessment to the manuscript. **Changes made:** p. 3.

6) Several risk thresholds are presented, but the underlying approaches' reliability information is limited. Since the authors frequently refer to risk stratification before even evaluating the model findings, the reliability of the approach to pre-test stratification of risk is important.

R: As suggested by Reviewer 1, we addressed this by extending the information concerning the clinical use of the respective thresholds as presented in reply to comment 1. **Changes made:** p. 7.

7) The authors state (line 154): “Finally, we combine predictions from both the statistical model and deep learning approach with the cardiologist’s post-test judgment by training a new logistic regression model on all three scores from the training set. We refer to this collaborative approach”. The collaborative model, touted as being the most successful, is unclear to the reviewer. How is this post-test judgment quantified? How is it combined numerically?

R: As suggested by Reviewer 1, we clarified the collaborative approach in more detail. The post-test VAS by the cardiologist (estimated probability between 0-100% after stress testing) was combined with the scores of the deep learning approach and the conventional machine learning model. These three scores were used to train a logistic regression model to predict the ground truth label. The logistic regression was exclusively trained on training set predictions and labels to prevent overfitting. **Changes made:** p. 3 and caption of Figure 1.

8) The 2-6-2 sequence of the stress ECG was constructed up to twenty times per patient for training the model. The authors should clarify the influence of

such an approach on application to a real-world setting. How was this applied in the validation setting?

R: Indeed, in the external validation set, we did not have access to the stress test segmentation. However, having access to heart rate measurements, we approximated the end of the stress test phase as the time at which the maximum heart rate was reached. Since we extract the 6 seconds from the stress phase from the end backwards, we capture the same physiological signals as in the training set. Sampling PRE and RECOVERY phase does not require any methodological adjustments.

9) The paper is somewhat dense, and many aspects of the modeling and decisions are not well explained. This leads to a challenge in understanding the value of some of the findings when they are presented.

R: As suggested by Reviewer 1, we reworked all sections and figure captions and highlighted the most relevant insights more precisely to provide a better understanding of the findings.

10) The tables and figures are similarly not adequately positioned, and the reviewer was challenged in finding the content essential to interpreting the study, If the set-up of the study and the order of presentation was better presented, I believe the readership will benefit substantially.

R: As suggested by Reviewer 1, we have moved Figure 1 to appear after the section on model development and Figure 4 to the end of the section to enhance clarity and facilitate easier interpretation of the study for readers. We will optimize the other positions in the journal's layouting step.

Reviewer #2 (Remarks to the Author):

This paper is addressing an early detection technique for Functionally relevant coronary artery disease (fCAD) that can result in premature death or nonfatal acute myocardial infarction. Two machine learning (ML) techniques were used to predict fCAD: initial one is using a small number of static clinical data, whereas the later one leverages electrocardiogram (ECG) signals from exercise stress testing. The results demonstrate that ML can significantly outperform cardiologists in predicting the presence of stress-induced fCAD in terms of area under the receiver operating characteristics.

The paper is well written and provide proper justification of the methods used. The dataset (3522 patients for ECG) is very important feature of this study and it is properly validated by cardiologist.

Machine Learning techniques used in the paper are conventional ML techniques that are used in different applications.

Overall, paper is good and provide good framework how to complement cardiologist by using this framework.

My only criticism with the results presents for younger age or less than 60 years of age population. As the dataset' median value of age is 66.7 years with standard deviation is 11.1 that how you can predict model for lesser age.

R: As suggested by Reviewer 2, we added more detailed information concerning the age distribution of the patients to the manuscript. We added a paragraph to the limitation section clarifying that in the derivation cohort 26% of the patients were younger than 60 years and 7.6% were younger than 50 years of age. **Changes made:** p. 19.

In Fig. 2, Sensitivity and Precision were shown, It is good idea to present the Specificity, Recall and Accuracy and F1 score will presented as well. Overall it is good paper.

R: As suggested by Reviewer 2, we added this information to the manuscript. We added Extended Data Table 4 which includes PPV, specificity, F1 score and accuracy metrics.

Reviewer #3 (Remarks to the Author):

SUMMARY

Bock et al. present the results of a retrospective analysis evaluating the predictive performance of machine learning for detection of functionally significant coronary artery disease. The authors demonstrate that machine learning can outperform cardiologist estimations of the risk of functionally significant CAD, potentially decreasing the number of patients that would require further risk stratification. Additionally, there seemed to be benefit from combining ML predictions with cardiologist estimates.

GENERAL COMMENTS FOR THE AUTHORS

The authors have performed a comprehensive study which includes model development and external testing for predicting 'functionally relevant' coronary artery disease. This could potentially help physicians choose between functional or anatomic imaging approaches. The most novel aspect of the model is that it incorporates stress ECG data automatically, which can be challenging clinically given the high frequency of noisy baseline data. However, the importance of this is minimized from a clinical perspective because the stress ECG is typically collected during the functional imaging test (instead of performing an exercise stress prior to deciding on which imaging test to pursue). The model more accurately predicts the presence of functionally relevant CAD compared to

expert clinical judgement with stress ECG results. There are a few similar studies which have been performed with machine learning (Most similar study, DOI: 10.1007/s12350-022-03012-6; internal testing n=20,418 from 5 sites; external testing n=9,019 from two sites; AUC 0.76 in external testing); (doi: 10.1155/2021/3551756; n=2503); (<https://link.springer.com/article/10.1007/s12350-018-1304-x>); (doi: 10.2196/16975); (DOI: 10.1016/j.cvdhj.2022.02.002). The first study has larger and more diverse training and external testing populations. The authors present another finding of significant interest; integrating expert judgement with the ML prediction improved performance (although there are issues with the current methodology as outlined in point 3 below).

A major concern is that the primary outcome, 'functionally relevant' CAD is a combination of two outcomes (myocardial ischemia by SPECT or obstructive CAD on cath) without clear delineation of how many patients were classified with each method. Selective referral to invasive angiography induces some bias in the study since some clinical aspects which may be utilized by the ML model (ECG findings, symptoms) also influence decisions to pursue invasive angiography. Notably, the previously mentioned studies did not include hybrid outcomes such as the one used by Bock et al.

R: As suggested by Reviewer 3, we have provided data on the use of invasive angiography and its impact on diagnosing functionally relevant coronary artery disease. Importantly, we view the combination of these two modalities as a significant strength of our study. Given the broad range of pre-test probabilities among the enrolled patients, it is impossible to conduct invasive testing on everyone. The incorporation of invasive testing results whenever available further solidifies our gold standard because adjudication was based on a comprehensive set of information, not just imaging data alone. Notably, a positive perfusion scan was only overruled when coronary angiography revealed normal coronary arteries. Conversely, a negative perfusion scan was only overruled if coronary angiography detected a high-grade coronary lesion (exceeding 75%) or when the fractional flow reserve was below 0.80. **Changes made:** p. 3.

SPECIFIC COMMENTS FOR THE AUTHORS

1) The model is developed in a cohort of patients who were referred for SPECT MPI (which is typically intermediate or higher risk). This is not the same population as the screening population that the authors are discussing in the Introduction, limiting generalizability of the model.

R: As suggested by Reviewer 3, we have included more details regarding the pretest risk of the patients within the limitations section. During study enrollment, MPI-SPECT/CT was the standard non-invasive imaging modality at our institution and was applied to patients with a wide range of pre-test probability for CAD. To emphasize this, we added the adjusted CAD Consortium model

probabilities as used in the current ESC guidelines for diagnosis and management of chronic coronary artery disease, demonstrating that 29% of the included patients in the derivation and internal validation cohort had an estimated probability of <15%. **Changes made:** p. 18.

2) The definition of “functionally relevant” coronary artery disease outlined by the authors in the Introduction (symptoms of ischemia, death or MI) is not the same as the definition used in the study.

R: As suggested by Reviewer 3, we adapted the paragraph to clarify that functionally coronary artery disease in the study context refers to patients with symptoms caused by stable coronary artery disease. **Changes made:** p. 2.

3) The cardiologist judgement was combined with the ML prediction using a logistic regression model. Did the authors consider including it as a feature for the ML model instead? Additionally, why not provide the cardiologists with the results to see how this would change their post-test probability estimate (particularly relevant since this is how it would most likely be implemented)?

R: We did not consider adding the cardiologist’s judgement to the ML model as we wanted the ML models to be minimally biased towards human judgement. Our main goal with the usage of ML was to investigate the effectiveness of learning physiological patterns from both ECG and patient characteristics that are predictive of fCAD.

We wholeheartedly agree with the reviewer that studying our model’s impact on the cardiologist’s judgement to investigate how collaborative ML can shape clinical practice is vital. However, the focus of this investigation is to study the diagnostic value of ML in the prediction of fCAD which we consider a precursor to the prospective clinical studies that the reviewer proposes.

4) The authors utilized cardiologist judgement using a visual scale after the EST. What factors were the physicians using? Did any perform formal calculations of pre and post-test probability? Do you have any sense of inter-rater reproducibility for these estimates?

R: As suggested by Reviewer 3, we provided more information concerning the cardiologist’s judgement. The clinical judgement reflected clinical practice where some cardiologists might have used formal calculations and others solely relied on their clinical expertise. Nevertheless, we did not obtain data concerning this systematically. The estimate was provided by the treating physician at the time of presentation of the patient. We therefore cannot assess inter-rater reproducibility. **Changes made:** p. 3.

5) There are several available risk scores that could be used to predict the likelihood of obstructive CAD (updated Diamond Forrester, CAD consortium scores, etc.). Consider including comparisons to these scores as supplemental results.

R: As suggested by Reviewer 3, we added available risk scores including the CAD consortium scores (updated Diamond Forrester) and adjusted CAD consortium scores as currently used in the ESC guidelines to the manuscript and compared it to our models. The CAD Consortium score reached an AUROC of 0.65 (0.59-0.71) and the score used in the current ESC guidelines reached 0.68 (0.60-0.76) in the cohort without a CAD history. Both approaches result in lower AUROC & AUPRC than our proposed methods. **Changes made:** Figure 3 and Extended Data Table 2.

MINOR POINTS:

6) What proportion of patients underwent cath to adjudicate functionally relevant CAD? How many cases were “equivocal” on cath?

R: As suggested by Reviewer 3, we provided data concerning the usage of invasive angiography and the impact on the outcome functionally relevant coronary artery disease. **Changes made:** p. 3.

7) In the “Data collection, label generation, and robustness”, should state that the external population is also a SPECT MPI referral cohort.

We added this information in the respective section and expanded the section “External Validation Data” on p. 32.

8) The introduction should provide some mention of coronary artery calcium scoring or CCTA.

R: As suggested, we added a paragraph about CCTA and calcium scoring. **Changes made:** p. 2.

9) Consider explicitly stating the central adjudication of functionally significant CAD was done blinded to demographics and ECG stress data (assuming this is the case), if not discuss potential bias related to this.

R: As suggested by Reviewer 3, we added that the central adjudication was not blinded to demographics and ECG results. However, the final adjudication was based on imagining and invasive testing whenever available **Changes made:** p.3.

10) Consider removing unnecessary abbreviations (w.r.t. ; l.h.s. etc.)

R: As suggested, we removed unnecessary abbreviations.

11) It does not seem appropriate to ‘upsample’ patients with functionally relevant CAD in the external testing population (Page 15). This defeats the purpose of external testing (to see if the model works in other populations) by trying to make the populations more similar. Additionally, since you upsampled this population, it is misleading to say that the prevalence was 6% in Tables.

R: The purpose of upsampling the population is to make AUPRC scores easier to interpret. This can be helpful in understanding a model’s performance in the context of a random classifier whose AUPRC changes with the prevalence of the positive class. However, we do agree that Extended Data Table 3 was not consistent in its presentation. We now present the upsampled results in parentheses to put more focus on the results that are achieved on the original validation cohort. We also clarified this in the table’s caption.

12) What was the definition of positive TID ratio?

R: As suggested by Reviewer 3, we added the definition of a positive Transient ischemic dilation ratio (TID). Changes made: p. 32.

13) Consider including a comparison to the previous work discussed above

R: We thank the reviewer for pointing out the absence of two important references. First, in “Machine learning to predict abnormal myocardial perfusion from pre-test features”, Miller et al. train gradient-boosted decision trees on 26 features to predict the presence of abnormal myocardial perfusion as adjudicated by visual interpretation. A key difference between this study and the study of Miller et al. is the focus on a smaller set of eight static and easily obtainable clinical features, alongside the incorporation of ECG time series data in the deep learning model. Furthermore, the diagnostic gold standard included functional information (i.e. coronary angiography) whenever available rather than only imaging data. We have also conducted ablation studies to investigate the impact of different ECG leads and preprocessing methods which can be used as a starting point for future studies involving deep learning and ECG time series. Furthermore, we follow the recommendation to perform rigorous subcohort analyses vital to highlight our model’s strengths and weaknesses. We also perform an in-depth analysis of the clinical utility of our methods by reporting their predictive performance at thresholds recommended by European and American guidelines and by conducting a decision curve analysis. Finally, we perform interpretability analyses to underline the (dis)advantages of conventional ML and DL, thereby underlining Miller et al.’s observation of the importance of the “CAD history” variable. To summarize, our study focuses on

the clinical utility of conventional ML, deep learning, and their combination with human judgement to predict fCAD using static but also dynamic features (e.g., ECG signals) providing recommendations for ECG preprocessing, lead selection, and multi-task learning, culminating in competitive external, cross-modality validation performance (see updated Extended Data Table 3). Similarly, Megna et al. perform a comparison of different ML approaches trained on static clinical variables, not considering the ECG signal. Their study differs from ours in numerous aspects: 1) The authors excluded patients with a history of CAD, a population for which we demonstrate the benefit of a collaborative ML approach (mean AUROC of CARPE_{Coll.}: 0.68 vs. Cardiologist: 0.63), 2) no exercise stress testing is performed, the authors limit their analysis to pharmacologically induced stress limiting the application of their proposed method, and 3) no external validation is performed. Next to these differences, there are also similarities. For instance, our SHAP analysis confirms their reported importance of the “sex” and “age” variable for tree-based learners such as CARPE_{Clin.}.

Given that the first study requires substantially more variables than ours and the second one was performed on a small subcohort, we believe it best to refrain from a numerical comparison. However, we reference the studies in the appropriate sections.

REVIEWER COMMENTS

Reviewer #1 (Remarks to the Author):

Thanks to the authors for clarifying many of the questions I raised. I only have two minor points:

- I continue to be skeptical of the collaborative performance of clinicians and AI, which is posed as the best-performing model. Especially because in the absence of true blinding, clinicians in practice will likely defer to the model outputs over their own judgment ("automation bias"). I suspect this will drive the performance lower in practice.
- Since the subcohort analyses do not evaluate statistical interaction tests by different test types and subgroups, and do not account for multiple testing, some of the inferences on model performance differences are overstated.

Reviewer #2 (Remarks to the Author):

I am happy with the changes and paper looks far better than the original submission.

Reviewer #3 (Remarks to the Author):

SUMMARY

Bock et al. present the results of a retrospective analysis evaluating the predictive performance of machine learning for detection of functionally significant coronary artery disease. The authors demonstrate that machine learning can outperform cardiologist estimations of the risk of functionally significant CAD, potentially decreasing the number of patients that would require further risk stratification after clinical assessment. This is a first revision, and I reviewed the original manuscript.

GENERAL COMMENTS FOR THE AUTHORS

The authors have responded to the previous comments. Some responses to my comments were incomplete. However, I have limited the points which I think the authors still need to address to only the most critical issues. The unaddressed comments include:

- 1) Regarding the composite outcome for functional coronary artery disease, the authors have given the breakdown of how many patients underwent coronary angiography within 3 months (n=701) but should report model performance (and calibration) separately for patients with and without ICA.
- 2) The authors should include the fact the clinical judgement was integrated with the ML model using logistic regression is a limitation. It is unclear whether this would actually translate in clinical practice and tends to overestimate the utility of combined analysis.
- 3) How many cases does the following statement apply to?: "In case of equivocal findings from MPI-SPECT/CT and coronary angiography, and adjudication committee....."

I have one new comment:

- 4) Expert interpretation of MPI, presumably, was performed with knowledge of stress test results and clinical features (both available to the model). This may lead to overestimation of the performance of

stress ECG features in particular (and underscores the importance of the cath only analysis suggested in point 1).

Reviewer #1 (Remarks to the Author):

1. I continue to be skeptical of the collaborative performance of clinicians and AI, which is posed as the best-performing model. Especially because in the absence of true blinding, clinicians in practice will likely defer to the model outputs over their own judgment ("automation bias"). I suspect this will drive the performance lower in practice.
 - As suggested by Reviewer # 1, we added a respective paragraph to our limitation section.
2. Since the subcohort analyses do not evaluate statistical interaction tests by different test types and subgroups, and do not account for multiple testing, some of the inferences on model performance differences are overstated.
 - As suggested by Reviewer #1, we extended our Bonferroni correction by introducing an additional correction factor. The drawn conclusions remain largely the same. We furthermore added an interaction analysis in Supplementary Figure 4.

Reviewer #2 (Remarks to the Author):

I am happy with the changes and paper looks far better than the original submission.

Thank you for your positive feedback on our revisions; we greatly appreciate your acknowledgment that the paper looks far better than the original submission.

Reviewer #3 (Remarks to the Author):

1. Regarding the composite outcome for functional coronary artery disease, the authors have given the breakdown of how many patients underwent coronary angiography within 3 months (n=701) but should report model performance (and calibration) separately for patients with and without ICA.
 - As suggested by Reviewer #3, we performed the analyses in patients with and without coronary angiography assessment within 90 days. The model's discriminative performance increase remained consistent across patients (see Supplementary Figures 2 and 3).
2. The authors should include the fact the clinical judgement was integrated with the ML model using logistic regression is a limitation. It is unclear whether this would actually translate in clinical practice and tends to overestimate the utility of combined analysis.
 - As suggested by Reviewer #3, we added a respective paragraph to the limitation section.

3. How many cases does the following statement apply to?: "In case of equivocal findings from MPI-SPECT/CT and coronary angiography, and adjudication committee....."
 - As suggested by Reviewer #3, we added this information to the referenced paragraph. It applied to 147/701 or 21% of the 701 patients that underwent coronary angiography within 90 days.
4. Expert interpretation of MPI, presumably, was performed with knowledge of stress test results and clinical features (both available to the model). This may lead to overestimation of the performance of stress ECG features in particular (and underscores the importance of the cath only analysis suggested in point 1).
 - As suggested by Reviewer #3, we compared model performance in patients who underwent coronary angiography within 90 days and those who did not. The model's performance increase remained consistent across patients, regardless of whether they received invasive coronary artery assessment or not. We added a respective paragraph to the limitations.

REVIEWERS' COMMENTS

Reviewer #1 (Remarks to the Author):

Thanks for the revisions to the manuscript. I appreciate that there are limitations, and that authors have acknowledged them.

Reviewer #3 (Remarks to the Author):

SUMMARY

Bock et al. present the results of a retrospective analysis evaluating the predictive performance of machine learning for detection of functionally significant coronary artery disease. The authors demonstrate that machine learning can outperform Cardiologist estimations of the risk of functionally significant CAD, potentially decreasing the number of patients that would require further risk stratification. This is a second revision.

GENERAL COMMENTS FOR THE AUTHORS

The authors have made further revisions to the manuscript in response to the previous comments. The limitations continue to be the use of a composite definition of functionally significant CAD (abnormal SPECT or ICA findings), with adjudication of 21% of studies due to “equivocal findings”. This represents a significant proportion of studies. The authors should include details on how these were adjudicated.

The second limitation is that the combined model (ML plus clinician) is integrated using logistic regression, which does not translate into actual clinical utility.

Reviewer #3 (Remarks to the Author)

1. The limitations continue to be the use of a composite definition of functionally significant CAD (abnormal SPECT or ICA findings), with adjudication of 21% of studies due to “equivocal findings”. This represents a significant proportion of studies. The authors should include details on how these were adjudicated.
 - a. As suggested, we included details on how equivocal findings were adjudicated.
2. The second limitation is that the combined model (ML plus clinician) is integrated using logistic regression, which does not translate into actual clinical utility.
 - a. We revised the manuscript acknowledging the limitations of logistic regression where necessary.